# Valosin-containing protein (VCP/p97) inhibitors relieve Mitofusin-dependent mitochondrial defects due to VCP disease mutants

Ting Zhang[1], Prashant Mishra[2†], Bruce A Hay[2], David Chan[2], Ming Guo[1,3*]

[1]Department of Neurology, UCLA David Geffen School of Medicine, University of California, Los Angeles, United States; [2]Division of Biology and Biological Engineering, California Institute of Technology, Pasadena, United States; [3]Department of Molecular and Medical Pharmacology, UCLA David Geffen School of Medicine, University of California, Los Angeles, United States

**Abstract** Missense mutations of *valosin-containing protein* (*VCP*) cause an autosomal dominant disease known as inclusion body myopathy, Paget disease with frontotemporal dementia (IBMPFD) and other neurodegenerative disorders. The pathological mechanism of IBMPFD is not clear and there is no treatment. We show that endogenous VCP negatively regulates Mitofusin, which is required for outer mitochondrial membrane fusion. Because 90% of IBMPFD patients have myopathy, we generated an *in vivo* IBMPFD model in adult *Drosophila* muscle, which recapitulates disease pathologies. We show that common VCP disease mutants act as hyperactive alleles with respect to regulation of Mitofusin. Importantly, VCP inhibitors suppress mitochondrial defects, muscle tissue damage and cell death associated with IBMPFD models in *Drosophila*. These inhibitors also suppress mitochondrial fusion and respiratory defects in IBMPFD patient fibroblasts. These results suggest that VCP disease mutants cause IBMPFD through a gain-of-function mechanism, and that VCP inhibitors have therapeutic value.

**\*For correspondence:** mingfly@g.ucla.edu

**Present address:** [†]Children's Medical Center Research Institute, University of Texas Southwestern Medical Center, Dallas, United States

**Competing interests:** The authors declare that no competing interests exist.

## Introduction

IBMPFD is an autosomal dominant disease that afflicts multiple body systems (*Kimonis et al., 2008a*). 90% of IBMPFD patients display skeletal muscle weakness (myopathy), the primary and earliest symptom of IBMPFD (*Weihl et al., 2009*). With progression of the disease, 50% of patients will develop Paget's disease of bone, affecting the skull, spine, hips and long bones of all four limbs. One-third of the patients will also develop frontotemporal dementia (*Weihl et al., 2009*; *Kimonis et al., 2008b*). Single missense mutations of *p97/cdc48/Valosin-containing protein (VCP)* cause fully penetrant IBMPFD (*Watts et al., 2004*). *VCP* mutations are also associated with 1–2% of cases of amyotrophic lateral sclerosis (ALS), familial hereditary spastic paraplegia (HSP) and Charcot-Marie-Tooth 2 (CMT2) disease (*Abramzon et al., 2012*; *de Bot et al., 2012*; *Gonzalez et al., 2014*).

*VCP* encodes a highly conserved and abundant AAA+ ATPase which participates in multiple cellular processes (*Meyer et al., 2012*). Human p97/VCP and its *Drosophila* homologue have 85% identity and 93% similarity in protein sequence. VCP has three major domains: the regulatory N domain, and the D1 and D2 ATPase domains. VCP hexameric rings utilize the energy from ATP hydrolysis to promote protein and RNA homeostasis, often by directly or indirectly modifying the fate of ubiquitin-labeled proteins (*Meyer et al., 2012*). VCP functions in multiple contexts that include protein quality control in the endoplasmic reticulum (*Ye et al., 2001*; *Shih and Hsueh, 2016*), chromatin

**eLife digest** A disease called "inclusion body myopathy, Paget disease and frontotemporal dementia (IBMPFD)" is an inherited disorder that can affect the muscles, brain and bones. People affected by the disease find that their muscles become progressively weaker, and may go on to develop a bone disorder and a form of dementia. The disease is caused by mutations in a gene that codes for Valosin-Containing Protein (VCP) – a protein that carries out many different roles in cells. Mutated forms of VCP predominantly affect tissues and organs that need a lot of energy, such as the muscles and the brain.

Within cells, structures called mitochondria generate energy. A number of studies suggest that the mitochondria in cells taken from individuals with IBMPFD do not generate energy properly. However, it is not known how mutant VCP disrupts mitochondria or how this leads to disease.

Fruit flies and humans have similar versions of the gene that produces VCP. Studying flies that have mutations that affect the gene can therefore help researchers to understand how these mutations might affect humans. Zhang et al. have now engineered fruit flies whose muscle cells made a mutant form of VCP. These flies showed many of the symptoms of IBMPFD (such as the death of muscle cells). In addition, the mitochondria in the muscle cells were smaller and more fragmented than normal. This led Zhang et al. to look at a protein called Mitofusin that controls how mitochondria fuse. VCP normally degrades Mitofusin; the mutant form of VCP caused Mitofusin to degrade excessively. This resulted in mitochondria experiencing reduced levels of fusion, leading to cell malfunction and death.

In further experiments, Zhang et al. treated the disease-modeled flies and cells from human patients with IBMPFD with inhibitor drugs that prevent the activity of VCP. This treatment reversed the defects that affected the mitochondria and prevented the death of muscle cells. This opens up the possibility of using VCP inhibitors – which are already being investigated in clinical trials as a treatment for cancer – as drugs to treat IBMPFD.

modification (*Puumalainen et al., 2014*; *Dobrynin et al., 2011*; *Vaz et al., 2013*), endolysosomal sorting (*Ritz et al., 2011*), membrane fusion (*Zhang et al., 2014*), autophagosome/lysosome function (*Ju et al., 2009*; *Johnson et al., 2015*), ER protein translocation (*DeLaBarre et al., 2006*; *Weihl et al., 2006*), formation of stress granules (*Buchan et al., 2013*) and ciliogenesis (*Raman et al., 2015*). VCP interacts with a number of co-factors to regulate these processes (*Meyer and Weihl, 2014*; *Meyer et al., 2012*), making it challenging to identify the molecular basis of phenotypes associated with disease mutations.

Disease-causing, single missense mutations of VCP are mainly located in the N-terminal half of the protein, either in the N domain or the D1 domain. Among them, the R155H mutation is the most frequently identified in IBMPFD patients, while the A232E mutation is associated with the most severe clinical manifestation (*Kimonis et al., 2008a*; *Ritson et al., 2010*). *In vitro* assays show that disease mutants have enhanced ATPase activity (*Weihl et al., 2006*; *Zhang et al., 2015*; *Niwa et al., 2012*; *Manno et al., 2010*; *Tang and Xia, 2013*). However, because VCP assembles as a hexamer, it is controversial whether disease mutants with increased ATPase activity cause disease through a dominant-active (*Chang et al., 2011*) or dominant-negative mechanism (*Ritz et al., 2011*; *Ju et al., 2009*; *Kim et al., 2013*; *Bartolome et al., 2013*; *Kimura et al., 2013*).

VCP disease mutants predominantly affect organs that have a high level of energy expenditure, such as brain and muscle. Mitochondria provide the bulk of the ATP to these tissues through oxidative phosphorylation, and mitochondrial functional defects, including mitochondrial uncoupling and decreased ATP production, are observed in IBMPFD patient fibroblasts (*Bartolome et al., 2013*; *Nalbandian et al., 2015a*). Abnormal mitochondria are also observed in transgenic VCP disease mutant R155H mice as well as VCP R155H knock-in mice (*Custer et al., 2010*; *Nalbandian et al., 2012*). These observations suggest that mitochondrial dysfunction is important for the pathogenesis of IBMPFD, but the mechanism by which VCP mutation alters mitochondrial function is not clear.

Mitochondrial morphology is controlled by dynamic cycles of fusion, controlled by Mitofusin (Mfn), and fission, regulated by DRP1 (*Chan, 2012*). Recent studies have uncovered the roles of

mitochondria fusion and fission defects in the pathogenesis of multiple neurodegenerative disorders (*Davies et al., 2007*; *Chen et al., 2003*; *Wakabayashi et al., 2009*), particularly Parkinson's disease, the second most common neurodegenerative disorder (*Guo, 2012*; *Pickrell and Youle, 2015*; *Deng et al., 2008*; *Yang et al., 2008*; *Poole et al., 2008*, *2010*; *Park et al., 2009*). In mammals, homologous proteins Mitofusin 1 and 2 (Mfn1 and Mfn2) mediate mitochondrial outer membrane fusion, with loss of function of Mfn 1 and 2 resulting in fragmented mitochondria and multiple defects in mitochondrial function (*Chen et al., 2003b*). In Hela cells, VCP promotes Mfn 1 degradation (*Xu et al., 2011*). VCP also mediates Mfn 1 and 2 degradation when mitophagy is stimulated in mammalian cells, and overexpression of VCP in *Drosophila* leads to downregulation of a tagged Mfn-transgene (*Kim et al., 2013*; *Kimura et al., 2013*). These observations led us to investigate the mitochondrial basis and molecular mechanisms for VCP disease mutants function using both *Drosophila* and IBMPFD patient cell models. As IBMPFD show the highest penetrance in muscle, with 90% of patients manifesting phenotypes in this tissue, we generated *Drosophila* models of IBMPFD in muscle, which recapitulate disease pathologies. We provide evidence in both *Drosophila* and human patient cells that VCP disease mutants have an enhanced ability to promote Mfn degradation, loss of which is associated with defects in mitochondrial fusion and physiology. Consistent with the hypothesis that VCP disease phenotypes are due to increased activity on substrates, we find that VCP ATPase activity inhibitors such as NMS-873 and ML240 can significantly rescue mitochondrial defects, disrupted muscle integrity and muscle cell death *in vivo* in *Drosophila*, and mitochondrial fusion and respiration defects in IBMPFD patient fibroblasts.

## Results

### Endogenous VCP regulates mitochondrial fusion via negative regulation of Mfn protein levels

Since VCP disease mutants have dramatic muscle phenotypes in humans and mouse models, and these are associated with defects in mitochondrial structure and function, we first examined the consequences of manipulating VCP levels. The *Drosophila* adult indirect flight muscle (IFM) is a non-essential, post-mitotic, and energy-intense tissue containing a high density of mitochondria. It also shows strong and consistent phenotypes in response to altered expression of genes required for mitochondrial fusion, fission and quality control (*Clark et al., 2006*; *Deng et al., 2008*; *Yang et al., 2008*; *Poole et al., 2008*; *Park et al., 2009*; *Poole et al., 2010*; *Yun et al., 2014*). Using the UAS-Gal4 system (*Brand and Perrimon, 1993*), we expressed VCP under the control of an IFM promoter derived from the *flightin* gene we previously generated (*Yun et al., 2014*). While expression of VCP under the control of the pan-muscle Gal4 drivers 24B-Gal4 or Mef2-Gal4 resulted in 100% adult lethality, expression of wildtype VCP under IFM control, which provides a pulse of expression in late pupal stages and early adulthood, gave rise to viable flies with intact and healthy muscle following adult eclosion. We utilized mitochondrially targeted GFP (mitoGFP) as a mitochondrial marker, as well as transmission electron microscopy (EM) at the ultrastructual levels for enhanced resolution particularly for cristae morphology. In 2-day-old flies, expression of VCP results in muscle with small mitochondria with intact cristae (*Figure 1—figure supplement 1A–B'*). Similar phenotypes are also observed in muscle from 6-day-old VCP-expressing flies (*Figure 1A–B''*). Conversely, expression of UAS-VCP RNAi under IFM control (VCP RNAi flies) results in mitochondria with an elongated phenotype (*Figure 1C–C''*, *Figure 1—figure supplement 1C and C'*). Two independent VCP RNAi lines were utilized and both showed significant knockdown (*Figure 1—figure supplement 2*).

To explore the generality of VCP's ability to regulate mitochondrial morphology we also examined mitochondrial morphology phenotypes of a *vcp* loss-of-function mutation. *vcp* null mutants in *Drosophila* are embryonic lethal (*Ruden et al., 2000*). The indirect flight muscle is a cellular syncytium and therefore cannot be used for mosaic analysis. As an alternative, we focused on the *Drosophila* female germline, in which a *vcp* loss-of-function mutant can be monitored in individual nurse cells, which are relatively large and offer great resolution for examining mitochondrial morphology. $vcp^{K15502}$ is a strong loss-of-function allele that has a P element insertion at the 5'-UTR of *Drosophila VCP* gene (*Chang et al., 2011*; *Ruden et al., 2000*) and has been used to generate *vcp* loss-of-function clones in imaginal discs (*Zhang et al., 2013*). The *Drosophila* ovary consists of numerous developing egg chambers. Each egg chamber develops from stage 1 to stage 14 and has

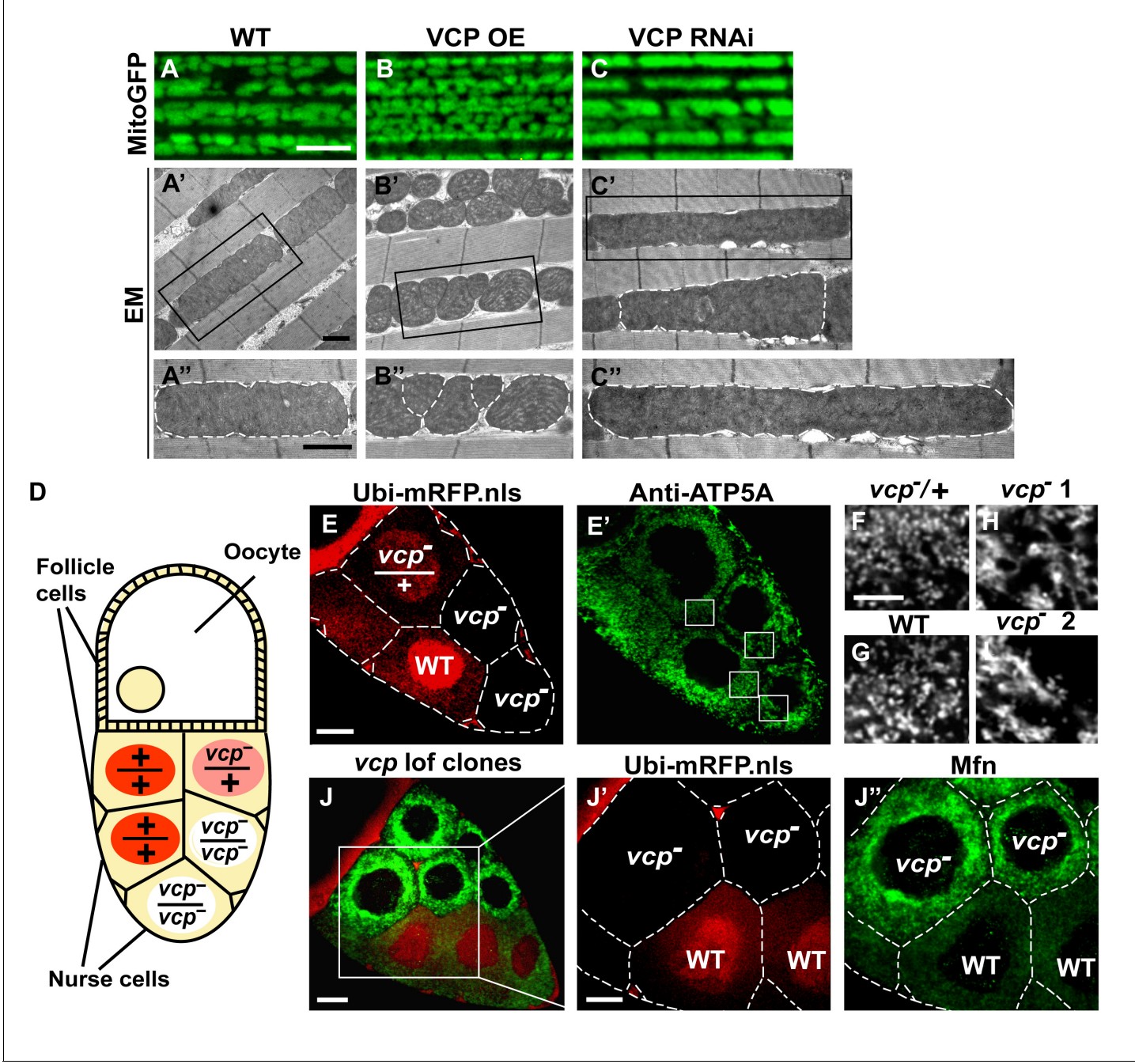

**Figure 1.** Endogenous VCP regulates mitochondrial fusion via negative regulation of Mfn protein levels. (A–C) MitoGFP localization to mitochondria serves as a marker for healthy mitochondria and their morphology in the indirect flight muscle. VCP overexpression (VCP OE, B) results in small mitochondria compared to wildtype (WT) flies (A); VCP RNAi expression results in elongated mitochondria (C). Scale Bar: 5 µm. (A'–C'') Electronic microscopic (EM) images show that VCP OE (B'–B'') results in smaller mitochondria as compared with WT (A'–A''). VCP RNAi generates elongated mitochondria with intact cristae (outlined with white dashed lines in C' and C''). (A'–C') Lower magnification; (A''–C'') Higher magnification of mitochondria outlined with black solid lines in A'–C'. Scale bar: 1 µm. (D): Schematic diagram of nurse cell mosaic analysis in *Drosophila* female germline. A stage 10A egg chamber has three types of cells. The monolayer follicle cells coat the surface of the egg chamber; 15 nurse cells provide proteins and RNAs to the oocyte during oogenesis. Heat shock induction of mitotic recombination during larvae stages results in the creation of a mosaic pattern in the adult nurse cells. Ubi-mRFP.NLS (Red) is used as a clone marker. Wildtype (WT) cells are labeled with two copies of RFP, +/+; heterozygous mutant cells have one copy of RFP and one copy of *vcp* loss-of-function mutant, *vcp*$^{K15502}$/+; homozygous mutant cells are RFP negative and have two copies of *vcp* loss-of-function mutant, *vcp*$^{K15502}$/*vcp*$^{K15502}$. (E): A stage 10A egg chamber with a mosaic pattern of nurse cells. Red signal is Ubi-mRFP.NLS. Scale bar: 20 µm. (E') Anti-ATP5A antibody staining is used to visualize mitochondrial morphology in nurse cells. Scale bar: 20 µm. (F–I) Higher magnification view of mitochondrial morphology in E' (outlined in white solid lines). Mitochondria appear as discrete and punctate structures

*Figure 1 continued on next page*

Figure 1 continued

in wildtype (**G**) and heterozygous *vcp* mutant cells (**F**), but becomes elongated and clumped in homozygous *vcp* loss-of-function mutant cells (**H and I**). Scale bar: 5 µm. (**J**) A stage 10B egg chamber with a mosaic pattern of nurse cells. Red signal is Ubi-mRFP.NLS; Green signal is anti-GFP staining of pCasper-Mfn-eGFP. Scale bar: 20 µm. (**J′–J′′**) Higher magnification of the egg chamber in J (outlined in white solid lines). Wildtype cells are RFP positive and homozygous *vcp* loss-of-function mutant cells are RFP negative (**J′**). pCasper-Mfn-eGFP levels significantly increase in homozygous *vcp* loss-of-function mutant cells (**J′′**). Scale bar: 10 µm.

The following figure supplements are available for figure 1:

**Figure supplement 1.** VCP overexpression leads to smaller mitochondria, while VCP RNAi leads to elongated mitochondria in 2-day-old indirect flight muscles.

**Figure supplement 2.** IFM-Gal4 driven UAS-VCP RNAi results in a significant decrease in VCP levels.

**Figure supplement 3.** pCasper-Mfn-eGFP colocalizes with the mitochondria marker ATP5A, and is silenced by expression of dsRNA targeting endogenous Mfn.

three types of cells: follicle cells, nurse cells and the oocyte (*Frydman and Spradling, 2001*). The germline complement of each egg chamber (15 nurse cells and the oocyte) derives from four sequential divisions of a single daughter of a stem cell. We utilized the FLP/FRT system (*Theodosiou and Xu, 1998*) to create mitotic recombinants in which nurse cells were either wildtype (+/+), heterozygous ($vcp^{K15502}$/+) or homozygous ($vcp^{K15502}/vcp^{K15502}$) for the *vcp* loss-of-function mutation (*Figure 1D*). In this system wildtype and heterozygous *vcp* loss-of-function mutant cells carry 2 or 1 copies of ubi-mRFP.NLS (RFP) respectively, whereas the homozygous *vcp* mutant cells are RFP negative (*Figure 1E*). Anti-ATP5A antibody is used to visualize mitochondrial morphology in nurse cells (*Figure 1E′*). Mitochondria in wildtype and heterozygous *vcp* mutant cells have a punctate morphology (*Figure 1F and G*). In contrast, in homozygous *vcp* loss-of-function mutants mitochondria are more tubular and clumped (*Figure 1H and I*). Together, these results suggest that endogenous VCP negatively regulates mitochondrial fusion or positively regulates fission.

Next, we investigated how *vcp* loss-of-function leads to elongated mitochondria. First, we examined the expression levels of Mfn in the female germline, using mosaic analysis, as above. We used a tagged genomic rescue transgene (pCasper-Mfn-eGFP, a transgene construct with a GFP tagged Mfn under the control of its endogenous promoter, a kind gift from Dr. CK Yao) to monitor the endogenous Mfn level as the existing anti-Mfn antibody lacked sufficient sensitivity. As expected, the genomic rescue Mfn-eGFP signal colocalizes with the mitochondria marker ATP5A in the nurse cells (*Figure 1—figure supplement 3A*). This signal is eliminated following Mfn RNAi, indicating that genomic rescue Mfn-eGFP is a reliable marker for endogenous Mfn (*Figure 1—figure supplement 3B*). In each egg chamber, *vcp* loss-of-function cells are adjacent to wildtype cells, providing an ideal opportunity to unambigously compare the levels of Mfn. As shown in *Figure 1J–J′′*, nurse cells homozygous for a *vcp* loss-of-function mutant have greatly increased Mfn levels compared to wildtype and heterozygous *vcp* loss-of-function cells. The findings that endogenous VCP regulates Mfn *in vivo* are unexpected and important, as VCP is a highly abundant cytosolic protein with many targets and VCP does not show appreciable localization to mitochondria in unstressed cells.

Second, we examined whether alteration of *vcp* can regulate Mfn levels in tissue lysate. Overexpression of wildtype VCP (VCP OE) in the flight muscle resulted in a decrease in endogenous Mfn levels (*Figure 2A*). VCP overexpression also caused a decrease in the levels of Mfn when Mfn was overexpressed (Mfn OE) using the IFM promoter, suggesting that VCP regulates Mfn post transcriptionally (*Figure 2A*). Conversely, VCP RNAi resulted in an increase in endogenous Mfn levels in a wildtype background, and an increase in total Mfn levels in the presence of Mfn OE (*Figure 2B*). In contrast, VCP overexpression does not alter the levels of pro-fission protein DRP1 (*Figure 2C*). In addition, VCP's regulation of Mfn is specific and not part of a general mitophagy response since the levels of other mitochondrial proteins, such as the outer membrane protein Porin, the inner membrane protein NDUSF3, and the matrix protein MnSOD, are not altered (*Figure 2D*).

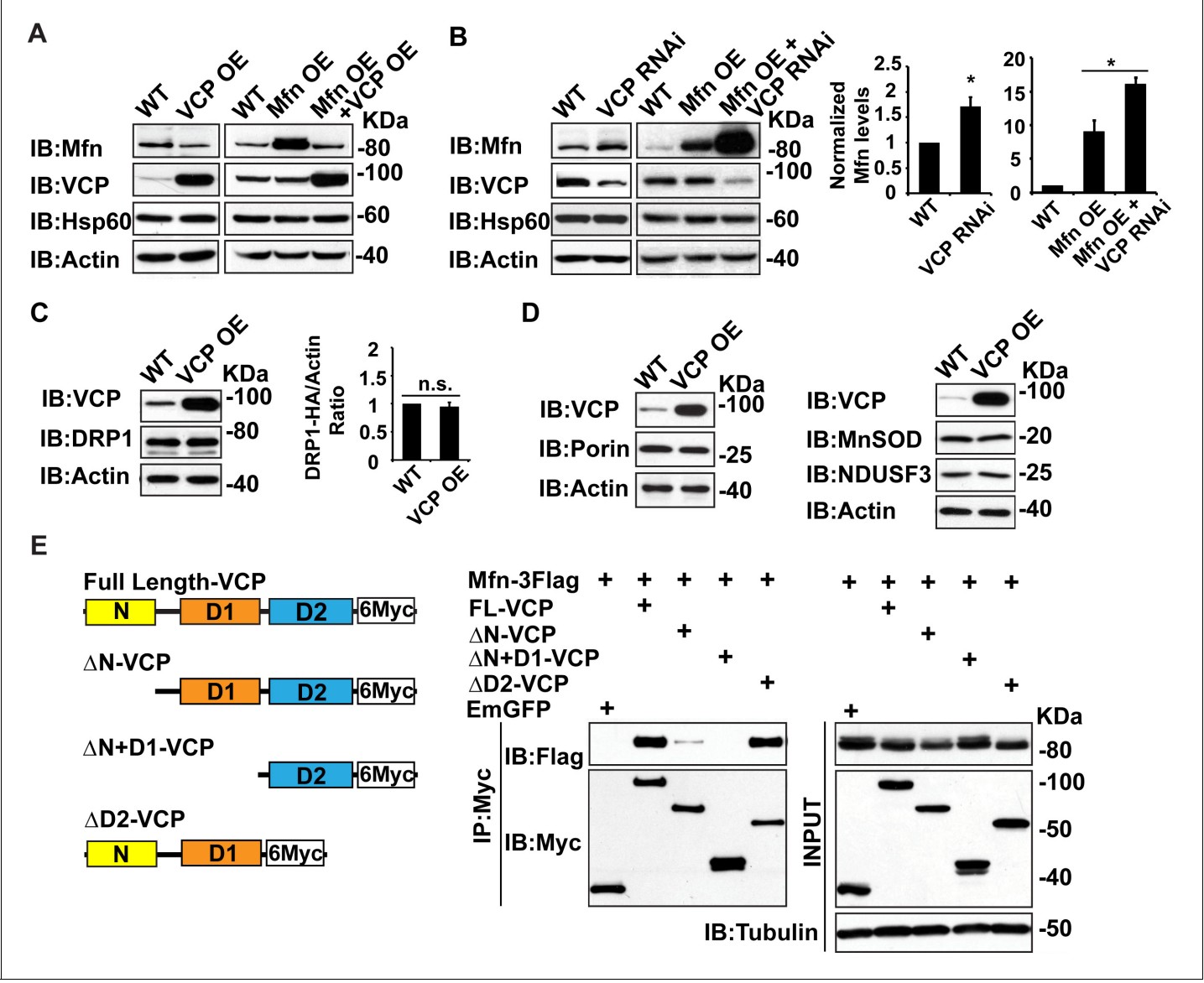

**Figure 2.** Mfn is a specific target of VCP. (A) In fly thoraxes, VCP OE in wildtype (IFM-Gal4 control) and Mfn OE background results in a significant decrease in Mfn levels. Hsp60 was used as a mitochondria control; Actin was used as a loading control. (B) VCP RNAi leads to enhanced Mfn levels (1.72 ± 0.37 compared to wildtype, set as 1; p=0.021, independent t-test, N = 4). VCP RNAi results in accumulation of Mfn in the Mfn OE background (16.15 ± 0.85 as compared to Mfn OE alone, 9.02 ± 1.63, wildtype set as 1, p=0.0101, independent t test, N = 3). *p<0.05. Mfn levels are normalized with Hsp60 and are displayed as mean±½SD. (C) VCP OE also does not result in changes in the levels of the pro-fission protein DRP1 level (p=0.088, independent t test, N = 3). DRP1 was detected using an HA tagged genomic rescue transgene expressed under the control of the endogenous DRP1 promoter (pCasper-DRP1-HA). n.s.: no statistical difference. (D) Markers for various mitochondrial compartments, including Porin (outer membrane), MnSOD (matrix), and NDUSF3 (inner membrane) were not altered by VCP overexpression. (E) Schematic diagram of full length and truncated VCP forms used in protein interaction assays in S2 cells. Protein interactions are assayed between full length and truncated VCP forms and Mfn. N and D1 domains are essential domains for the VCP-Mfn interaction.

## VCP physically interacts with Mfn via the N and D1 domains

To explore the possibility that Mfn is a direct target for VCP, we asked if Mfn could physically interact with VCP in immunoprecipitation assays from S2 cells. Indeed, full length VCP strongly interacts with Mfn (*Figure 2E*). This is consistent with previous findings (*Kim et al., 2013*). VCP has been shown to interact with substrate through the D1 ATPase domain, with the presence of the N domain being essential for the interaction (*Meyer et al., 2012*). As shown in *Figure 2E* , this is also the case

with the VCP-Mfn interaction. When the N domain of VCP is deleted, the strength of VCP-Mfn interaction is decreased, and it is undetectable when N and D1 are both deleted. In contrast, removal of the D2 domain has no effect on interaction between VCP and Mfn. Interestingly, Mfn levels in the input blot are decreased under conditions when VCP and Mfn interact, but not under conditions in which interactions are not observed, consistent with the hypothesis that direct interaction between VCP and Mfn is required for Mfn degradation. Together, these observations suggest that Mfn is a specific, direct target of VCP.

## VCP suppresses *PINK1/parkin* mutant mitochondrial phenotypes in muscle

Mutations in *PINK1* and *Parkin* lead to autosomal recessive forms of Parkinson's disease (PD) (*Kitada et al., 1998*; *Valente et al., 2004*). We and others have shown that *PINK1* and *Parkin* function in the same pathway, with *PINK1* positively regulating *Parkin* to control mitochondrial integrity and quality control in *Drosophila* (*Clark et al., 2006*; *Yang et al., 2006*; *Park et al., 2006*). Importantly, levels of Mfn are significantly increased in *PINK1* and *parkin* null mutants and downregulation of Mfn protein levels suppresses phenotypes in both mutants (*Deng et al., 2008*; *Yang et al., 2008*; *Park et al., 2009*; *Yu et al., 2011*). The results presented above suggest that expression of VCP, by decreasing the level of Mfn, could also suppress *PINK1/parkin* phenotypes. Kim et al. have argued that VCP expression suppresses *PINK1*, but not *parkin* loss-of-function phenotypes. This, along with their observation that VCP levels at mitochondria increased following expression of Parkin in the presence of CCCP, suggested a model in which VCP action on mitochondria requires recruitment by Parkin (*Kim et al., 2013*). Because the EDTP-Gal4 driver used in these earlier experiments does not express at significant levels in flight muscle (*Seroude et al., 2002*) (*Figure 3J*, see GFP signals), we reexamined this issue. We expressed VCP in the flight muscle of *PINK1* and *parkin* mutants. As previously shown (*Yang et al., 2006*; *Clark et al., 2006*; *Park et al., 2006*), *PINK1* and *parkin* null mutants show mitochondrial defects and tissue disintegration with vacuolation in muscle (*Figure 3A–B'' and D–D''*). At the ultrastructural level, *PINK1* and *parkin* mutants display swollen mitochondria with a broken cristae (*Figure 3A''', B''' and D''*). VCP OE completely suppressed tissue damage and mitochondrial defects (*Figure 3C–C''' and E–E'''*). The thorax indentation phenotype in *PINK1/parkin* mutants is also completely suppressed by VCP overexpression under the control of the IFM-Gal4 driver, but not the EDTP-Gal4 driver (*Figure 3—figure supplement 1*). The suppression observed by Kim, et al. is probably due to non-disjunction of the *PINK1* null flies. As predicted for a mechanism of VCP action that occurs through regulation of Mfn, VCP OE significantly reduced the accumulation of Mfn normally observed in both mutants (*Figure 3F*).

To further substantiate the finding that overexpression of *VCP* suppresses *parkin* null mutants, we also examined VCP's ability to suppress mitochondrial phenotypes associated with loss of both *parkin* and *mul1*. MUL1 (MULAN/MAPL) is a RING-containing E3 ligase that acts as a negative regulator of Mfn in both *Drosophila* and mammals (*Yun et al., 2014*). When overexpressed, it suppresses *PINK1* and *parkin* null mutant phenotypes by degrading Mfn, while loss of *mul1* leads to increased levels of Mfn. Thus, *parkin mul1* double null mutants have strikingly more severe mitochondrial phenotypes than those observed in flies lacking *parkin* alone (*Yun et al., 2014*). Remarkably, expression of VCP resulted in a dramatic rescue of mitochondrial morphology due to lack of both *parkin* and *mul1* (*Figure 3G–I'*). Together, these results suggest that VCP-dependent suppression of *PINK1* null, *parkin* null and *parkin mul1* double phenotypes occurs through downregulaton of Mfn levels (*Figure 3K*). Together, our observations argue that VCP regulates Mfn levels under non-stress conditions. They also show that VCP-dependent regulation of Mfn does not require PINK1 or Parkin. However, these results do not exclude the possibility that Parkin activation further promotes VCP-dependent Mfn degradation under some situations.

## Expression of VCP disease mutants in muscle recapitulates pathological features of IBMPFD

To explore the basis of myopathy induced by VCP disease mutants we sought to create a model of IBMPFD in *Drosophila* muscle, as 90% of the IBMPFD patients show muscle phenotypes. Since *VCP* disease mutations are autosomal dominant, we characterized the consequences of disease mutant overexpression. The pathologies observed in IBMPFD patient muscle, and in muscle from VCP

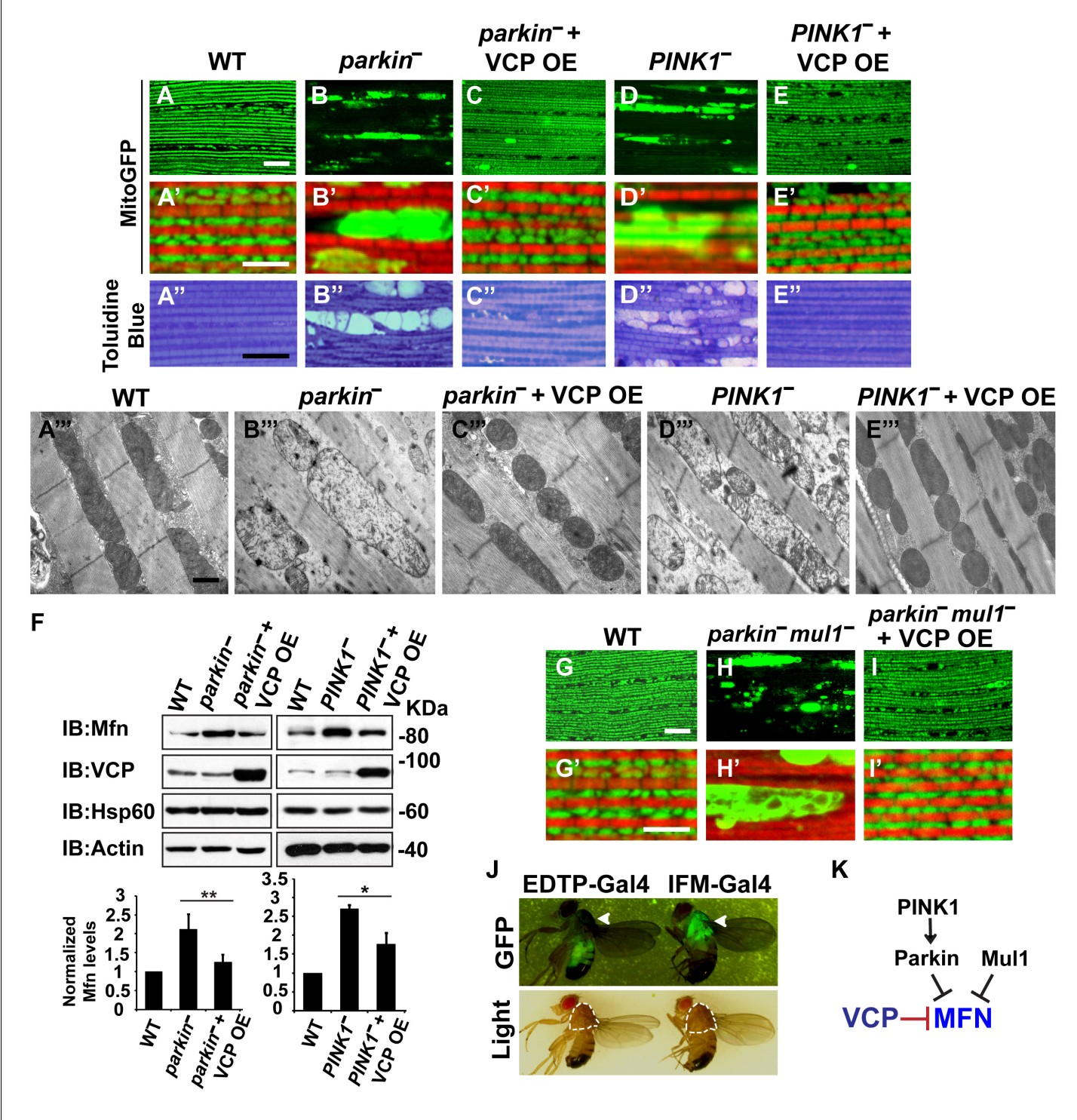

**Figure 3.** VCP overexpression suppresses mitochondrial defects in *PINK1* null, *parkin* null and *parkin mul1* double null mutants. (**A–E'**) Compared to wildtype (**A** and **A'**), *parkin* and *PINK1* mutants lose MitoGFP signal and accumulate large aggregates. VCP OE (IFM-Gal4>UAS-VCP) significantly rescues the MitoGFP phenotype in both mutants. Filamentous actin is stained with Rhodamine Phalloidin (Red). (**A–E**) Lower magnification. Scale bar: 20 µm. (**A'–E'**) Higher magnification. Scale bar: 5 µm. (**A''–E''**) Toluidine Blue shows that vacuole formation in the muscle tissue in *parkin* and *PINK1* mutants is robustly suppressed by VCP OE. Thoraxes are assayed 2 days after eclosion. Scale bar: 30 µm. (**A'''–E'''**) At the ultrastructural level, wildtype (WT, IFM-Gal4 control) mitochondria are well aligned with compact cristae (**A'''**). *parkin* and *PINK1* mutants display swollen mitochondria with broken cristae (**B'''** and **D'''**). VCP OE (IFM-Gal4>UAS-VCP) completely rescued the mitochondrial defects in both mutants (**C'''** and **E'''**). Thoraxes are sectioned at 2 days after eclosion. Scale bar: 1 µm. (**F**) VCP OE caused a decrease in Mfn accumulation in *parkin* and *PINK1* mutants. Mfn protein levels

*Figure 3 continued on next page*

*Figure 3 continued*

in the *parkin* mutant increased to 2.12 ± 0.69 as compared with wildtype (set as 1). VCP OE caused a decrease in Mfn levels in the *parkin* mutant to 1.26 ± 0.34 as compared with *parkin* (p=0.005, independent t test, N = 3. **p<0.01). In *PINK1* mutants, Mfn levels increased to 2.69 ± 0.11 as compared with wildtype, while VCP OE in *PINK1* mutant caused Mfn levels to decrease (1.77 ± 0.29 as compared with *PINK1*, p=0.023, independent t test, N = 3, *p<0.05). Normalized Mfn levels are shown as mean±½SD. (**G–I'**) Compared to wildtype, the *parkin mul1* mutant lacks the most MitoGFP signal and large aggregates are present (**G–H'**). VCP OE (IFM-Gal4>UAS-VCP) rescues the MitoGFP phenotype in *parkin mul1* mutants (**I** and **I'**). Myofibrils are stained with Rhodamine Phalloidin (Red). (**G–I**) Lower Magnification, Scale bar: 20 μm; (**G'–I'**) Higher Magnification, Scale bar: 5 μm. (**J**) Expression pattern of EDTP-Gal4 and IFM-Gal4 in 3-day-old flies. EDTP-Gal4 driven UAS-MitoGFP signal is barely detected in the thorax, where the indirect flight muscle is located, suggesting that EDTP-Gal4 does not have sufficient flight muscle specific expression. MitoGFP is present at high levels in the thoraxes of IFM-Gal4 driven flies. Thorax structure is outlined with dashed white lines in white light and indicated with a white arrowhead in GFP field. (**K**) Schematic diagram showing that PINK/Parkin in parallel with Mul1 negatively regulates Mfn protein levels; VCP negatively regulates Mfn protein levels independent of these modifiers.

The following figure supplement is available for figure 3:

**Figure supplement 1.** IFM-Gal4 driven UAS-VCP expression but not EDTP-Gal4 driven UAS-VCP rescues thorax defects in *PINK1* and *parkin* mutants in *Drosophila*.

disease mutant transgenic and mouse knock-in models, include muscle cell death, changes in mitochondrial morphology, damaged tissue integrity, TAR DNA-binding protein 43 (TDP43) mislocalization to the cytosol, and formation of autophagic marker p62 and ubiquitin aggregates, signs of multisystem proteinopathy (*Ju et al., 2009*; *Ritson et al., 2010*; *Nalbandian et al., 2012*; *Ritz et al., 2011*; *Nalbandian et al., 2015b*; *Weihl, 2011*; *Ahmed et al., 2016*). Previous studies of VCP disease mutants in *Drosophila* muscle showed effects on tissue integrity and mitochondria cristae structure (*Chang et al., 2011*; *Kim et al., 2013*), but not other pathologies. We thus investigated the pathologies due to mutant VCP expression. We focused on expression of VCP R152H and A229E (corresponding to Human VCP R155H and A232E, hereafter referred to as VCP RH and VCP AE), as they are the most frequent and the most severe mutations identified, respectively (*Figure 4A*). Anti-VCP antibody blotting shows that VCP WT, VCP RH and VCP AE are expressed at comparable level in the fly's thorax (*Figure 4B*). In wildtype flies, flies with IFM-Gal4 insertion, and flies with VCP WT expression, the muscle structure was healthy and intact 6 days after eclosion. No cell death was observed and mitochondria were densely packed and contained high levels of MitoGFP (*Figure 4C, D and D'*). Muscles also displayed a well-aligned myofibril structure (*Figure 4E and E'*). Together these observations indicate that increased expression of VCP is not overtly toxic. In contrast, muscle from flies expressing VCP RH or VCP AE under IFM-Gal4 control (thereafter called VCP RH or VCP AE flies) showed extensive cell death (97 ± 3.6% and 95.7 ± 5.1% muscle are TUNEL-positive), defective MitoGFP signals (*Figure 4D'' and D'''*), and severely disrupted muscle integrity (*Figure 4E'' and E'''*). In addition, while in wildtype and VCP WT flies TDP43 was found in IFM nuclear and sarcoplasmic compartments, IFMs from VCP RH and AE flies showed a decrease in the intensity of the nuclear signal, and an increase in the intensity of staining associated with puncta in the sarcoplasmic area (*Figure 4C,F–F'''*). Muscle from VCP RH and AE flies also contained large aggregates of anti-p62 and anti-ubiquitin staining. These were not observed in wildtype or VCP WT flies (*Figure 4G–H'''*). Importantly, the pathologies observed in flight muscle at day 6 post eclosion were not present at day 2 (*Figure 4—figure supplement 1*), suggesting the degenerative nature of the pathology, as with the human disease and mouse models (*Kimonis et al., 2008a*; *Custer et al., 2010*; *Nalbandian et al., 2012*). Together, these observations indicate that IFM-specific expression of VCP RH and AE recapitulates a broad spectrum of IBMPFD disease pathologies and forms the strong basis for further investigation of disease mechanisms and treatment studies.

## VCP disease mutants are hyperactive with respect to downregulation of Mfn protein levels and inhibition of mitochondrial fusion

Next, we investigated the cellular basis of mitochondrial defects in our IBMPFD flies using electron microscopy. 6 days after eclosion, flies expressing VCP WT have smaller mitochondria as compared with control flies, but with intact cristae (*Figure 5A–B'*). In VCP RH and AE muscle, however, mitochondria are smaller in size, and more fragmented than in VCP WT flies. In addition, they also have

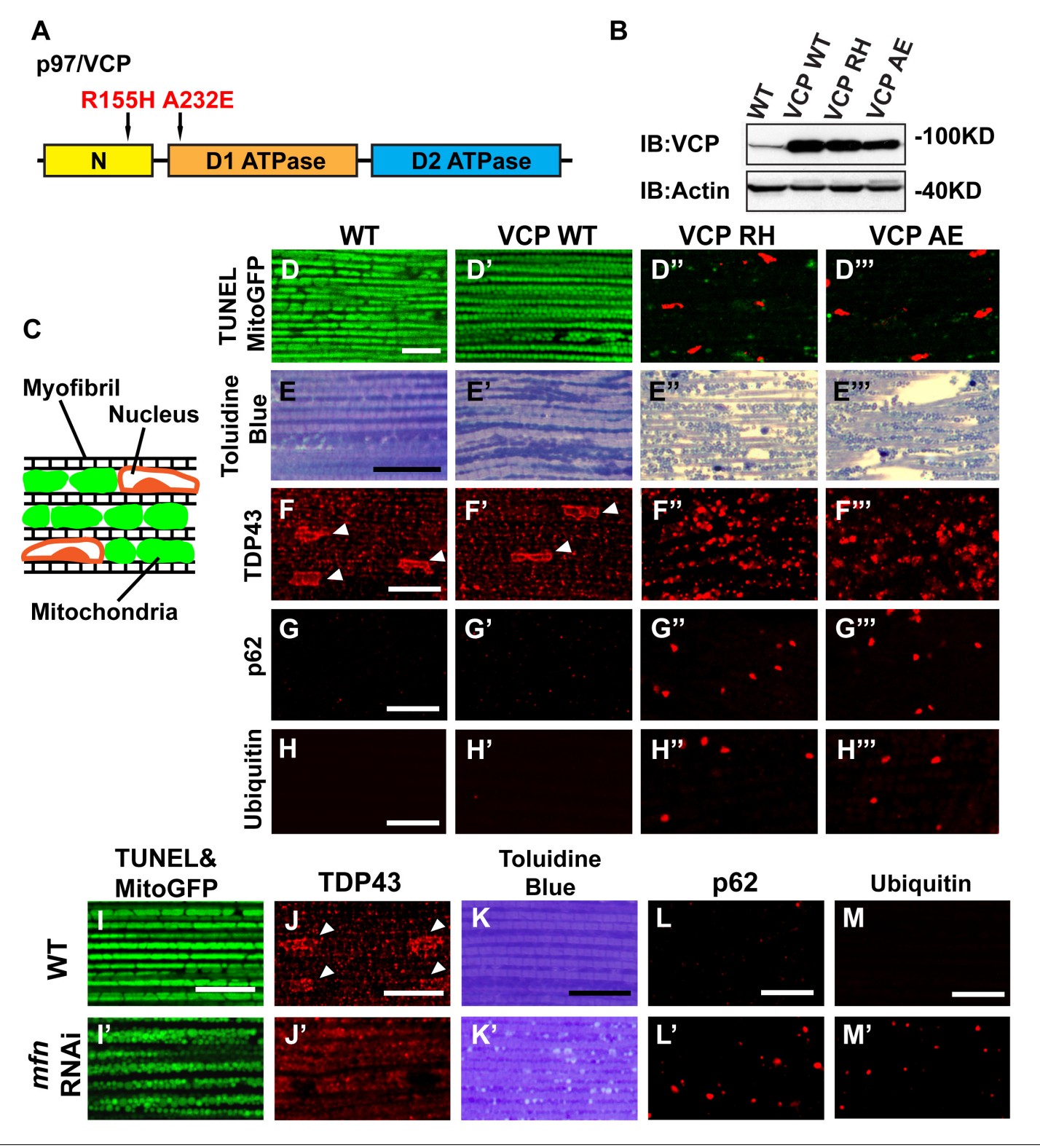

**Figure 4.** Expression of VCP disease mutants and *mfn* RNAi knocking down lead to pathology in adult muscle tissue. (**A**) Diagram of Human p97/VCP protein domains and two disease mutants, VCP R155H and A232E. Their corresponding *Drosophila* homologues are VCP R152H and A229E, hereafter referred to as VCP RH and AE. (**B**) Expression levels of UAS-VCP WT, RH and AE disease mutants expressed under IFM-Gal4 are comparable. (**C**) Diagram of *Drosophila* indirect flight muscle structure. Mitochondria (Green) and nuclei (Orange) are densely packed in between myofibrils (which contain large amounts of actin, Black). (**D–D'''**) MitoGFP (Green) and TUNEL staining (Red). 6-day-old WT (IFM-Gal4 control) flies and VCP WT flies have

*Figure 4 continued on next page*

*Figure 4 continued*

healthy muscles (no TUNEL staining) and are MitoGFP positive (D and D'). 6-day-old VCP RH and AE expressing flies show high levels of TUNEL staining. 97 ± 3.6% and 95.7 ± 5.1% of RH (p=0.03. RH v.s. WT, independent t test) and AE (p=0.015, AE v.s. WT, independent t test) and are MitoGFP negative (D'' and D'''). Scale bar: 10 μm. (E–E''') Toluidine Blue staining of muscle in WT, VCP WT, RH and AE flies. Myofibrils are well aligned with densely packed mitochondria in WT and VCP WT flies 6 days after eclosion (E and E'). In VCP RH and AE flies fiber structure is disrupted, and mitochondria are misaligned and lightly stained, with empty spaces in between (E'' and E'''). Scale bar: 40 μm. (F–F''') Anti-TDP43 antibody staining shows nuclear (white arrowhead) and sarcoplasmic localization of TDP-43 in WT and VCP WT adult fly muscle (F and F'). In VCP RH and AE flies the nuclear signal disappears and the signal is increased in muscle sarcoplasm (F'' and F'''). Scale bar: 5 μm. (G–G'') Anti-Ref(2)P/p62 antibody staining. The signal is weak and uniform in WT and VCP WT flies (G and G'), and punctate in VCP RH and AE flies (G'' and G'''). Scale bar: 5 μm. (H–H''') Anti-P4D1 ubiquitin antibody staining. Signal is weak and uniform in WT and VCP WT flies (H' and H'), and punctate in VCP RH and AE flies (H'' and H''''). Scale bar: 5 μm. (I–M) Effects of *mfn* RNAi in 8-day-old adult muscle tissue. (I–M) Wildtype muscle visualized with TUNEL and mitoGFP (I), anti-TDP43 (J), Toluidine blue (K), anti-p62 (L), and anti-ubiquitin (M). (I'–M') *mfn* RNAi muscle visualized with the same probes as above. Scale bar: 30 μm in K-K', 5 μm in the rest.

The following figure supplement is available for figure 4:

**Figure supplement 1.** Muscle isolated from flies expressing VCP disease mutants does not show gross defects at 2 days post eclosion.

severe ultrastructural defects under EM studies with a broken cristae (*Figure 5C–D'*). This phenotype is similar to that of flies with a muscle-specific Mfn knock down (*Figure 5E and E'*) (*Deng et al., 2008*). Muscle from 2-day-old VCP RH and VCP AE flies also show similar phenotypes to Mfn knock down (*Figure 5—figure supplement 1*).

The strong mitochondrial phenotypes observed in VCP RH or VCP AE predicts that expression of VCP RH or VCP AE may result in a greater decrease in Mfn levels as compared with the expression of VCP WT. This is indeed what we observed (*Figure 5F*). This raises the possibility that VCP RH and AE are hyperactive alleles. To further substantiate these findings, we compared the Mfn levels in flies expressing VCP E2Q, a well-established ATPase defective mutant containing two mutations (fly or human residues by number E305Q and E578Q) that abolish the ATPase activity of the D1 and D2 domains. Expression of E2Q did not significantly change the Mfn levels (*Figure 5F*). As a control, the expression of any of the above forms of VCP did not alter expression levels of mitochondrial matrix protein Hsp60 (*Figure 5F*). These results suggest that VCP disease mutants are hyperactive with respect to promoting downregulation of Mfn levels.

Based on these observations, we hypothesized that VCP disease mutants should also rescue mitochondrial phenotypes in *parkin* and *parkin mul1* double mutants. As shown in *Figure 5G–J*, expression of VCP RH and AE mutants, but not VCP E2Q or VCP RNAi, robustly restored mitochondrial structure, as indicated by MitoGFP fluorescence, in the *parkin* null mutant (*Figure 5K and L*). Moreover, corresponding to the mitochondrial phenotypes, the accumulation of Mfn in *parkin* mutants was significantly reduced in the presence of VCP WT, RH and AE, with VCP RH and AE causing greater decreases in Mfn levels than VCP WT (*Figure 5S*). In addition, VCP RH and AE expression also robustly rescued the mitochondrial defects in the *parkin mul1* double null mutants, but an expression of VCP E2Q or VCP RNAi did not (*Figure 5M–R*). Taken together, these results indicate that VCP RH and AE disease mutants do not behave as ATPase defective or loss-of-function mutants, and instead are hyperactive with respect to regulation of Mfn.

## Mfn downregulation is important to the muscle pathology in IBMPFD flies

A prediction from our finding that Mfn downregulation is important for IBMPFD pathology is that downregulation of Mfn should result in phenotypes similar to those observed with an expression of VCP disease mutants. Indeed, *mfn* muscle-specific knockdown flies have fragmented mitochondria (*Figure 4I–I'*), TDP43 mislocalization (*Figure 4J–J'*), vacuole formation (*Figure 4K–K'*), p62 and ubiquitin accumulation (*Figure 4L–L' and M–M'*), as with the VCP disease mutants. Next we asked if *mfn* overexpression could rescue the phenotypes due to VCP RH and AE. Since *mfn* overexpression results in significant mitochondrial phenotypes, tissue disintegration and cell death on its own (*Yun et al., 2014*), we are unable to address this question completely due to a technical limitation. We were, however, able to bring about a mild increase in Mfn levels through addition of 2 copies of

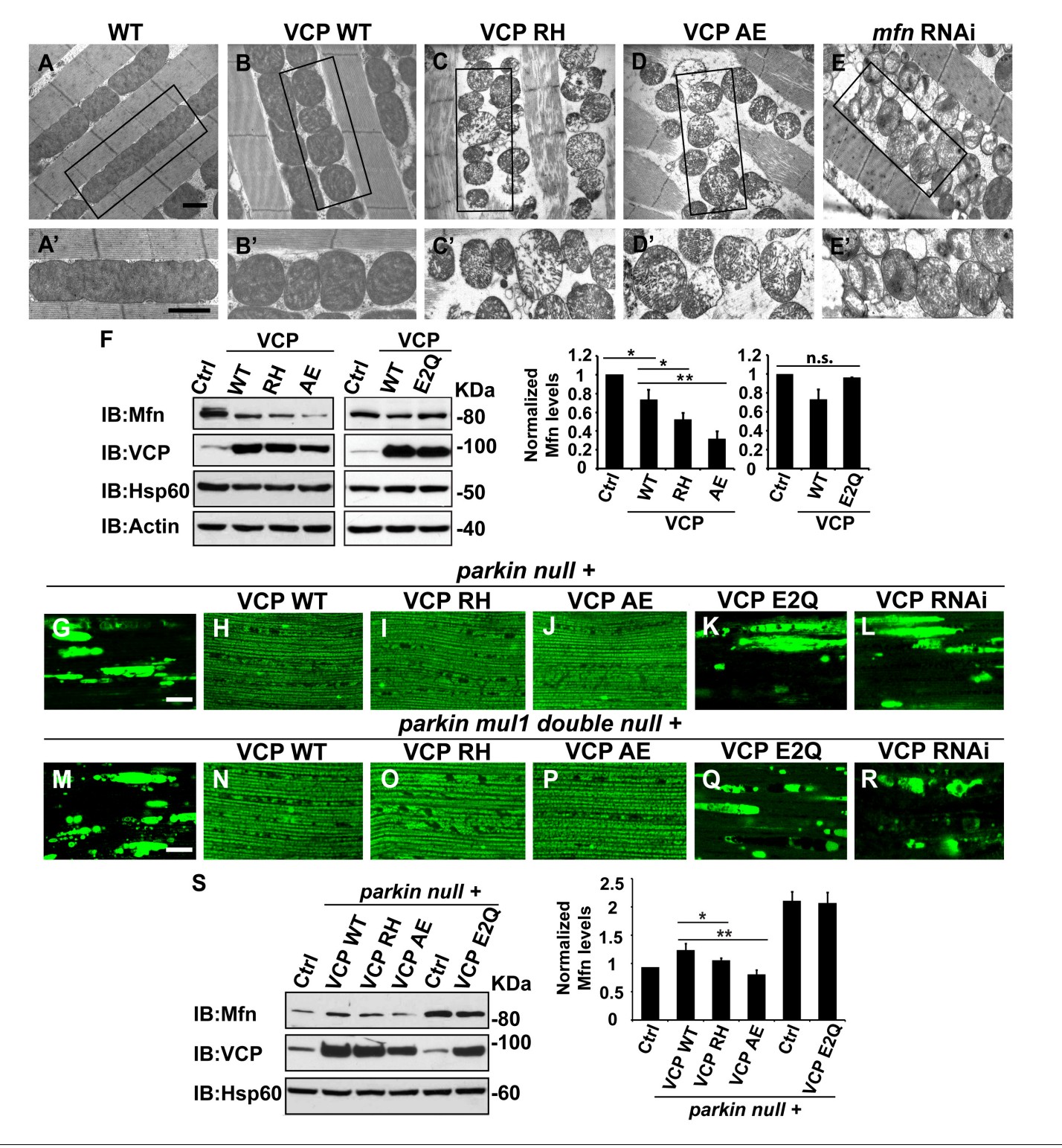

**Figure 5.** VCP disease mutants are hyperactive in downregulating Mfn protein levels and inhibiting mitochondrial fusion. (**A–B'**) Electronic microscopic images of muscle of different genotypes. Compared to WT (IFM-Gal4 control, **A** and **A'**), mitochondria are smaller with intact cristea in VCP WT (**B** and **B'**) 6 days after eclosion. Scale bar: 1 μm. (**C–E'**) Expression of VCP RH and AE leads to distorted fiber structure, and small mitochondria with broken or empty cristae. *mfn* RNAi flies have similar morphological defects in mitochondria (**E** and **E'**). Scale bar: 1 μm. (**F**) Expression of VCP RH and AE lead to a further reduction in Mfn as compared to VCP WT. Mfn levels in VCP WT are reduced to 0.73 ± 0.21 as compared with Gal4 control, which is set as 1 (p=0.037, independent t test, N = 3). Mfn levels in VCP RH and AE are reduced to 0.52 ± 0.14 (p=0.037, independent t test, N = 3) and 0.32 ± 0.16
*Figure 5 continued on next page*

*Figure 5 continued*

(p=0.003, independent t test, N = 3) as compared with Gal4 control. Mfn levels are normalized with those of Hsp60, a mitochondria marker. Expression of VCP ATPase defective mutant E2Q (p=0.097, independent t test, N = 3) does not significantly change the Mfn level in fly thoraxes as compared with Gal4 control. n.s.: no statistical significance, p>0.05. Normalized Mfn levels are shown as mean±½SD. (G–L) MitoGFP localization assay shows expression of VCP WT (H), RH (I) and AE (J) potently rescues the mitochondrial defects in *parkin* mutant (G). Expression of the ATPase defective mutant E2Q (K) or VCP RNAi (L) does not. Scale bar: 20 μm. (M–R) MitoGFP assay shows expression of VCP WT (N), RH (O) and AE (P) potently rescues the mitochondrial defects in *parkin mul1* double mutants (M). Expression of VCP E2Q (Q) and VCP RNAi (R) do not. Scale bar: 20 μm. (S) Western blot shows that increased Mfn levels normally present in a *parkin* null mutant are significantly decreased in VCP WT, RH and AE flies, but not in VCP E2Q flies. Mfn levels are further decreased in a *parkin* mutant expressing VCP RH (p=0.0228, independent t test, N = 3) or AE (p=0.00565, independent t test, N = 3), as compared to VCP WT. Samples are from 2-day-old fly thoraces. Normalized Mfn levels are shown as mean±½SD. *p<0.05; **p<0.01.

The following figure supplements are available for figure 5:

**Figure supplement 1.** Expression of VCP disease mutants leads to small mitochondria with abnormal cristae, phenotypes similar to *mfn* RNAi knocking down.

**Figure supplement 2.** Mfn expression can suppress mitochondrial defects observed in VCP RH and AE flies.

**Figure supplement 3.** Mfn expression does not significantly rescue the pathology in VCP RH and AE fly muscles.

---

an *mfn* genomic rescue transgene in the background of VCP RH and AE expressing flies. Interestingly, this resulted in a suppression of the mitochondrial fragmentation and cristae phenotypes in 2-day-old VCP RH and AE flies (*Figure 5—figure supplement 2*) but not the pathology observed in VCP RH and AE fly muscles at 6 days (*Figure 5—figure supplement 3*). Together, our data suggest that downregulation of Mfn is an important contributor to IBMPFD muscle pathology.

## IBMPFD patient fibroblasts with the VCP R155H mutation have mitochondrial respiratory chain and mitochondrial fusion defects

Next, we extended our studies in IBMPFD patient fibroblasts. We focused our studies on $VCP^{R155H/+}$ cells (thereafter called RH cells), as fibroblasts harboring the $VCP^{A232E/+}$ mutation are not available. First, we characterized mitochondrial respiration in control and immortalized patient fibroblasts. As shown in *Figure 6A*, the basal level Oxygen Consumption Rate (OCR) in the VCP RH patient cells is slightly decreased compared to the healthy control, suggesting that baseline mitochondrial respiratory chain function is compromised in patient cells. Maximum OCR, measured in the presence of the uncoupler FCCP, is significantly decreased in VCP RH IBMPFD patient cells. The spare OCR, which is calculated by subtracting the basal OCR from the maximum OCR, provides a measure of reserve mitochondrial capacity for ATP generation. The spare OCR in IBMPFD patient cells is significantly decreased as compared with controls. Others have recently reported similar observations (*Nalbandian et al., 2015a*).

Disruption of mitochondrial fusion can result in defects in respiration similar to those observed in patient cells (*Chen et al., 2003b*). Mfn 1 and 2 levels in VCP RH cells were reduced to 0.66 ± 0.14 and 0.60 ± 0.14 respectively, as compared with age and gender matched healthy control cells (set as 1, *Figure 6B*). To explore the functional significance of this decrease, we carried out mitochondrial fusion assays using fibroblasts from healthy controls and patients. Both cell populations were transfected with a plasmid encoding a mitochondrial matrix targeted, photoactivated GFP (PA-GFP). Following photoactivation of GFP within a region of the cell, the GFP fluorescence signal spreads and the intensity decreases as these mitochondria fuse with non-activated neighboring mitochondria. Thus, in this assay an increased rate of fluorescence loss corresponds to increased mitochondrial fusion activity (*Mishra et al., 2014a*). In control cells, PA-GFP gradually decreases after photoactivation; in the VCP RH IBMPFD patient cells, the GFP signal decrease is significantly delayed (*Figure 6C–D', E*). These data suggest that the VCP RH mutation results in a decrease in Mfn 1 and 2 levels that lead to impaired mitochondrial fusion. In addition, the VCP mutant impairs both basal and maximal mitochondrial respiratory function.

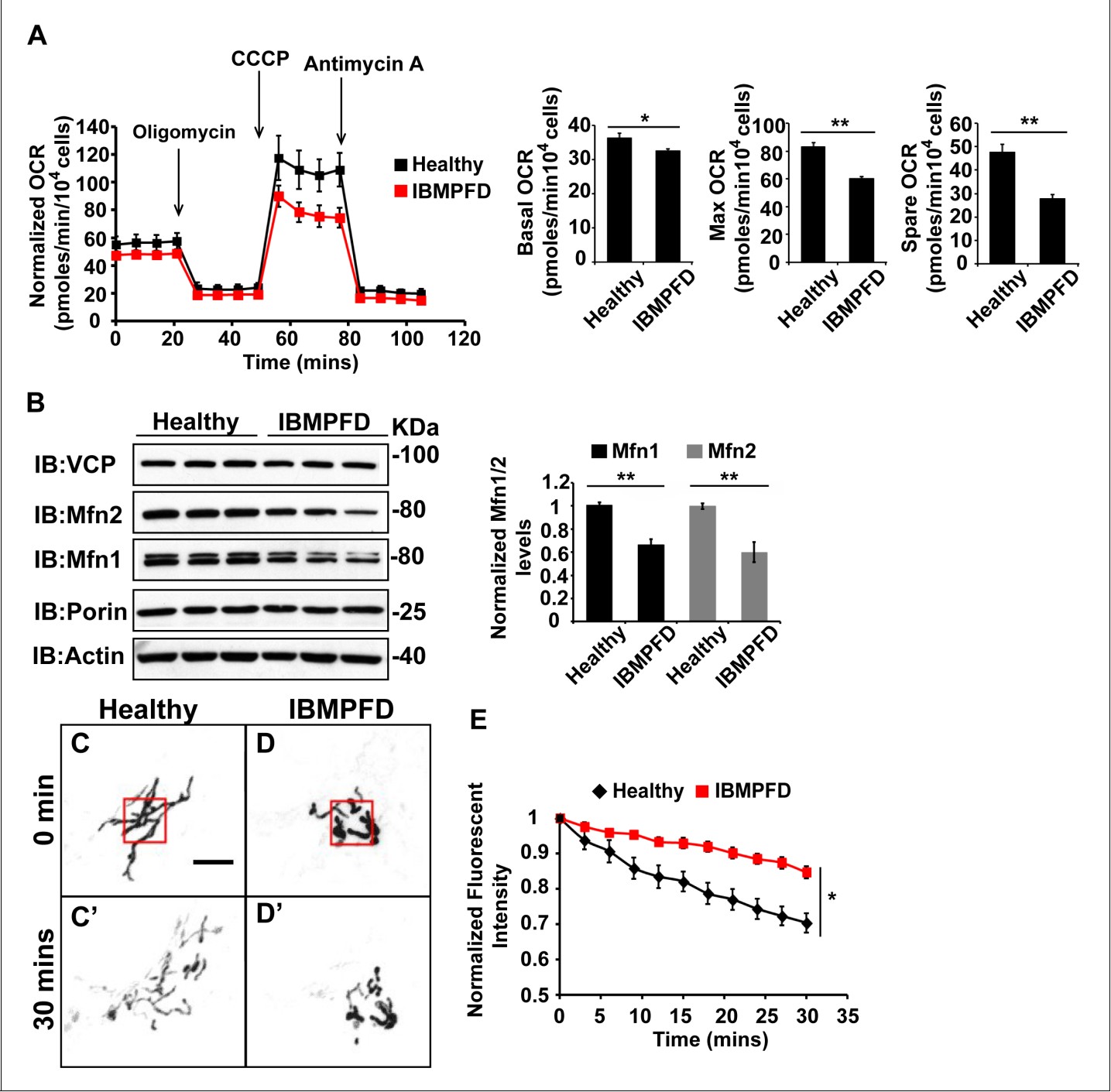

**Figure 6.** IBMPFD patient cells carrying the *VCP^R155H/+* mutation have decreased Mfn 1 and Mfn 2 levels and reduced mitochondrial fusion. (A) Oxygen consumption rates (OCR) in healthy control and IBMPFD patient fibroblasts. Inhibitory drugs were added at the time points indicated. Basal levels of OCR (p=0.026, independent t test, N = 8) and maximum (CCCP-stimulated) OCR (p=3.74E-06, independent t test, N = 8) are both decreased in the IBMPFD patient fibroblasts. The spare OCR is also significantly decreased (p=0.000024, independent t test, N = 8). (B) Mfn 1 and 2 levels are consistently decreased in three independent lysates from the IBMPFD patient fibroblasts when compared to the healthy control. Mfn 1 levels in patient fibroblasts are reduced to 0.66 ± 0.14 of the controls (p=0.001, independent t test, N = 5); Mfn 2 levels are reduced to 0.60 ± 0.14 of the controls (p=0.008, independent t test, N = 5). **p<0.01. Normalized Mfn 1 and 2 levels are shown as mean±½SD. (C–D') Mitochondrial fusion assay utilizing photoactivatable mitoGFP (PA-GFP) in healthy control and IBMPFD patient fibroblasts. After laser photoactivation of a region (Red hollow square in C and D), cells were tracked for 30 min. (E) PA-GFP signal intensity (throughout the whole cell) is recorded every 3 minutes, and the rate of decrease of average GFP signal is a measure of mitochondrial fusion. Fusion rates are significantly decreased in the IBMPFD patient fibroblasts when compared to the healthy control (C', D' and E, independent t test, *p<0.05).

# The VCP inhibitors NMS-873 and ML240 generate elongated mitochondria, block VCP disease mutant phenotypes, and VCP-dependent suppression of *PINK1* mutant mitochondrial defects

It has been controversial in the field whether VCP disease mutants with increased ATPase activity behave as dominant-active or dominant-negative mutants. Some studies show that VCP RH and AE, with enhanced ATPase activity (*Weihl et al., 2006*; *Zhang et al., 2015*; *Niwa et al., 2012*; *Manno et al., 2010*; *Tang and Xia, 2013*), result in hyperactivity in animal disease models (*Chang et al., 2011*). However, other cell-based studies suggest they behave in a dominant negative fashion (*Ritz et al., 2011*; *Ju et al., 2009*; *Kim et al., 2013*; *Bartolome et al., 2013*; *Kimura et al., 2013*). Understanding how VCP disease mutants behave is critical for understanding the disease process and finding possible therapies. We hypothesize that if an abnormally enhanced ATPase activity causes pathology, disease severity should ameliorate if ATPase activity is decreased through the use of VCP ATPase inhibitors. To test this hypothesis, we characterized the effects of VCP inhibitors *in vivo*.

Multiple VCP ATPase activity inhibitors have recently been described (*Chapman et al., 2015*). NMS-873, 3-[3-Cyclopentylsulfanyl-5-(4'-methanesulfonyl-2-methyl-biphenyl-4-yloxymethyl)-[1, 2, 4] triazol-4-yl]-pyridine, an allosteric inhibitor of VCP ATPase activity, is a highly specific and robust VCP ATPase inhibitor (*Magnaghi et al., 2013*). ML240, 2-(2-Amino-1H-benzimidazol-1-yl)−8-methoxy-N-(phenylmethyl)−4-quinazolinamine is another potent inhibitor, which competitively blocks ATP binding to VCP (*Chou et al., 2013*). These drugs and their derivatives were developed for treatment of cancers (*Magnaghi et al., 2013*; *Chou et al., 2013*; *Deshaies, 2014*; *Zhou et al., 2015*). The ability of these compounds to rescue defects in IBMPFD models has not been reported.

We first examined the effects of VCP inhibitors delivered to *Drosophila* through feeding. Various concentrations of drug or DMSO were included in food during the first to third larval instars, and muscle mitochondrial morphology was characterized at 2 and 6 days after eclosion. NMS-873 or ML240 feeding resulted in dramatic mitochondrial elongation in wildtype animals as visualized with MitoGFP using light microscopy (*Figure 7A–A''*) and at the ultrastructural level using EM (*Figure 7B–B'''*). These phenotypes are similar to those observed following VCP RNAi (*Figure 1C–C''*). In another respect, drug-treated animals developed normally (data not shown). The smaller mitochondria phenotype associated with overexpression of VCP WT was also significantly suppressed following feeding with 30 μM NMS-873 or 30 μM ML240 (*Figure 7C–C'''*). Feeding with NMS-873 and ML240 also reversed the phenotypic rescue of mitochondrial defects caused by expression of VCP WT, RH and AE in *PINK1* mutants (*Figure 7—figure supplement 1*). Supporting these results, feeding with NMS-873 also resulted in a dose-dependent increase in the levels of Mfn (*Figure 7D*). Together, these observations suggest that NMS-873 and ML240 inhibit endogenous VCP *in vivo*, resulting in a phenocopy of loss of VCP function.

We then fed NMS-873 and ML240 to VCP RH and AE disease mutant flies and analyzed mitochondrial phenotypes, muscle viability and integrity in 6-day-old flies. Flies were fed with 30 μM NMS-873 or 30 μM ML240 from first instar larvae until 6 days after eclosion. In control and VCP WT expressing flies, NMS-873 and ML240 feeding resulted in normal development to adulthood (data not shown) and did not cause muscle death or tissue damage; all muscles were TUNEL negative and had an intact tissue structure (*Figure 8—figure supplement 1*). Strikingly, NMS-873 and ML240 feeding largely prevented muscle cell death caused by expression of VCP RH or AE (*Figure 8A–B''*). It also significantly rescued disrupted muscle integrity (*Figure 8C–D''*). Rescue of mitochondrial size, cristae structure, and myofibril organization was also observed at the ultrastructural level (*Figure 8E–F''*). It is interesting to note that in the VCP disease mutants feeding groups (*Figure 8E', E'', F' and F''*), mitochondria remained somewhat smaller as compared to wildtype in *Figure 7B*. Elongated mitochondria, such as those observed in the wildtype and VCP WT feeding groups (*Figure 7B', B'', C' and C''*), were never observed in the VCP RH and VCP AE groups even though all were fed the same concentration of inhibitors. This partial suppression further suggests that these mutants are hyperactive. Finally, we note that VCP inhibitor feeding also resulted in significant rescue of a number of other phenotypes in VCP RH and AE expressing flies, including those associated with p62, ubiquitin and TDP43 (*Figure 8G–L''*). Together, these observations suggest that VCP disease mutants are hyperactive in multiple aspects, and that disease phenotypes can be suppressed by inhibition of VCP ATPase activity.

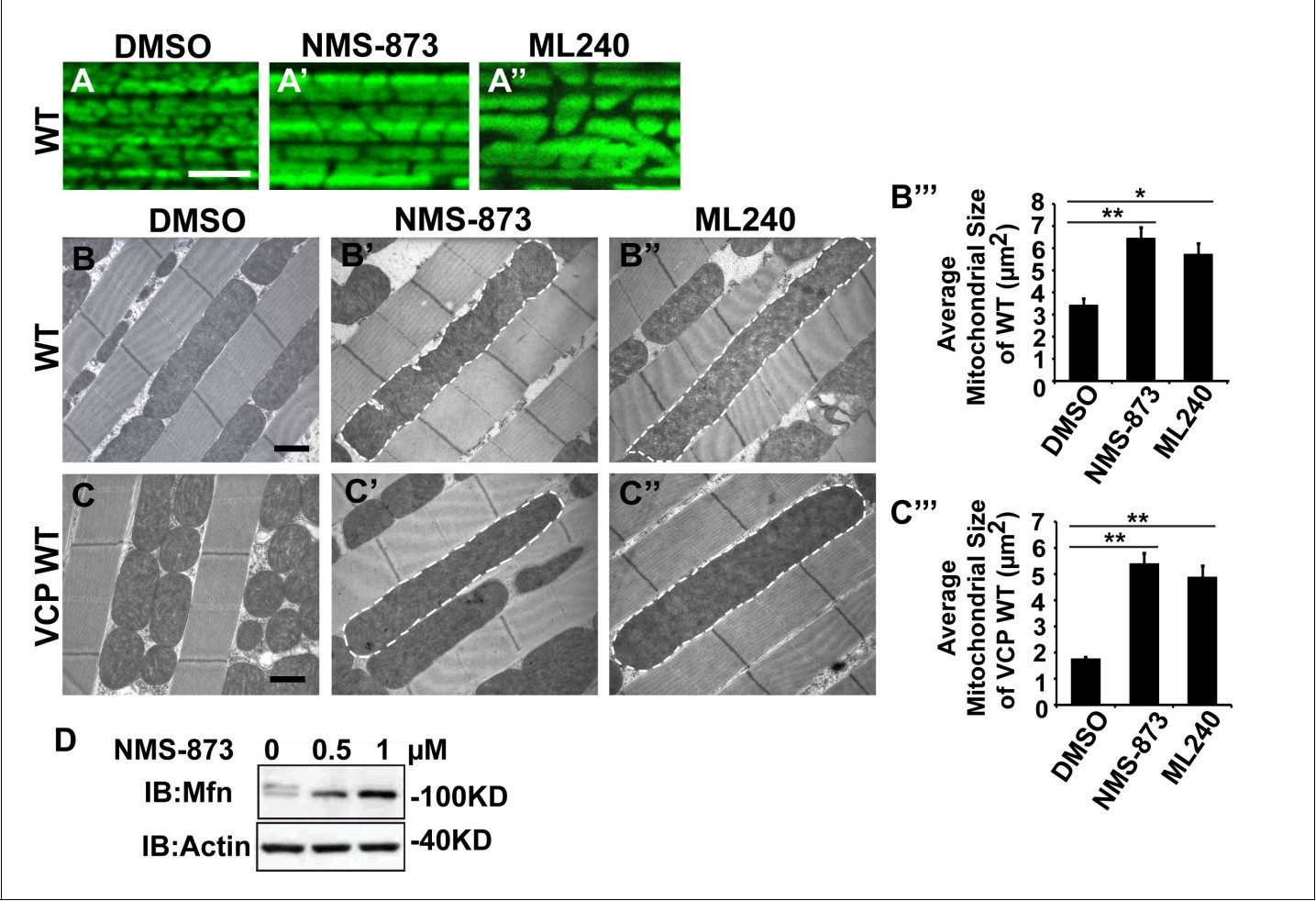

**Figure 7.** VCP inhibitors promote mitochondrial elongation. (**A–A''**) MitoGFP assay for mitochondrial morphology. Compared to DMSO-fed flies, 10 μM NMS-873 or ML240 feeding results in more fused mitochondria in 2-day-old flies. Scale bar: 5 μm. (**B–B''** and **C–C''**) In 6-day-old flies fed with 30 μM NMS-873 (**B'**) or 30 μM ML240 (**B''**), elongated mitochondria (outlined in dashed white lines) are observed in muscle. Mitochondria are small in VCP WT flies (**C**) as compared with WT (IFM-Gal4 control, **B**). This phenotype is reversed by 30 μM NMS-873 (**C'**) or 30 μM ML240 (**C''**) feeding and shifted towards a pro-fusion direction, as the mitochondria are also elongated as compared with WT (**B**). (**B'''** and **C'''**) Statistical analysis of mitochondrial size in EM cross sections. In wildtype flies, 30 μM NMS-873 or ML240 feeding results in a mitochondrial size increase to 6.46 ± 0.44 μm$^2$ (p=0.007, independent t test, N = 45) or 5.75 ± 0.48 μm$^2$ (p=0.032, independent t test, N = 39) as compared to the DMSO group (3.45 ± 0.27 μm$^2$, N = 47). VCP WT expression results in small mitochondria (1.78 ± 0.05 μm$^2$, N = 68) as compared with WT (**B**). Feeding of 30 μM NMS-873 (5.41 ± 0.40 μm$^2$, p=0.000, independent t test, N = 44) or ML240 (4.90 ± 0.42 μm$^2$, p=0.000, independent t test, N = 45) reverses these effects. Mitochondria size is shown as mean±½SEM. (**D**) pCasper-Mfn-eGFP levels accumulate in a dose dependent manner when treated with NMS-873 at 0.5 μM and 1 μM for 14 hr.

The following figure supplement is available for figure 7:

**Figure supplement 1.** VCP inhibitor treatment blocks VCP WT and disease mutant rescue of mitochondrial defects in a *PINK1* mutant.

## Low level of VCP inhibitor treatment suppresses mitochondrial fusion and respiration defects in IBMPFD patient's fibroblasts

The above data demonstrated a decrease in Mfn 1 and 2 levels, mitochondrial fusion defects and impaired mitochondrial respiratory function in IBMPFD patient cells. Next, we explored the effects of these drugs on these changes in IBMPFD patient fibroblasts. In patient cells, a high concentration of ML240 (10 μM, with IC50 of 110 nM) or NMS-873 (10 μM, with IC50 of 30 nM) led to cell death, and NMS-873 directly inhibits mitochondrial respiration (data not shown). However, when lower concentrations of ML240 are used, robust rescue effects are observed (*Figure 9*). As shown in *Figure 9A*, immortalized IBMPFD patient fibroblasts have decreased Mfn 1 levels, and these are

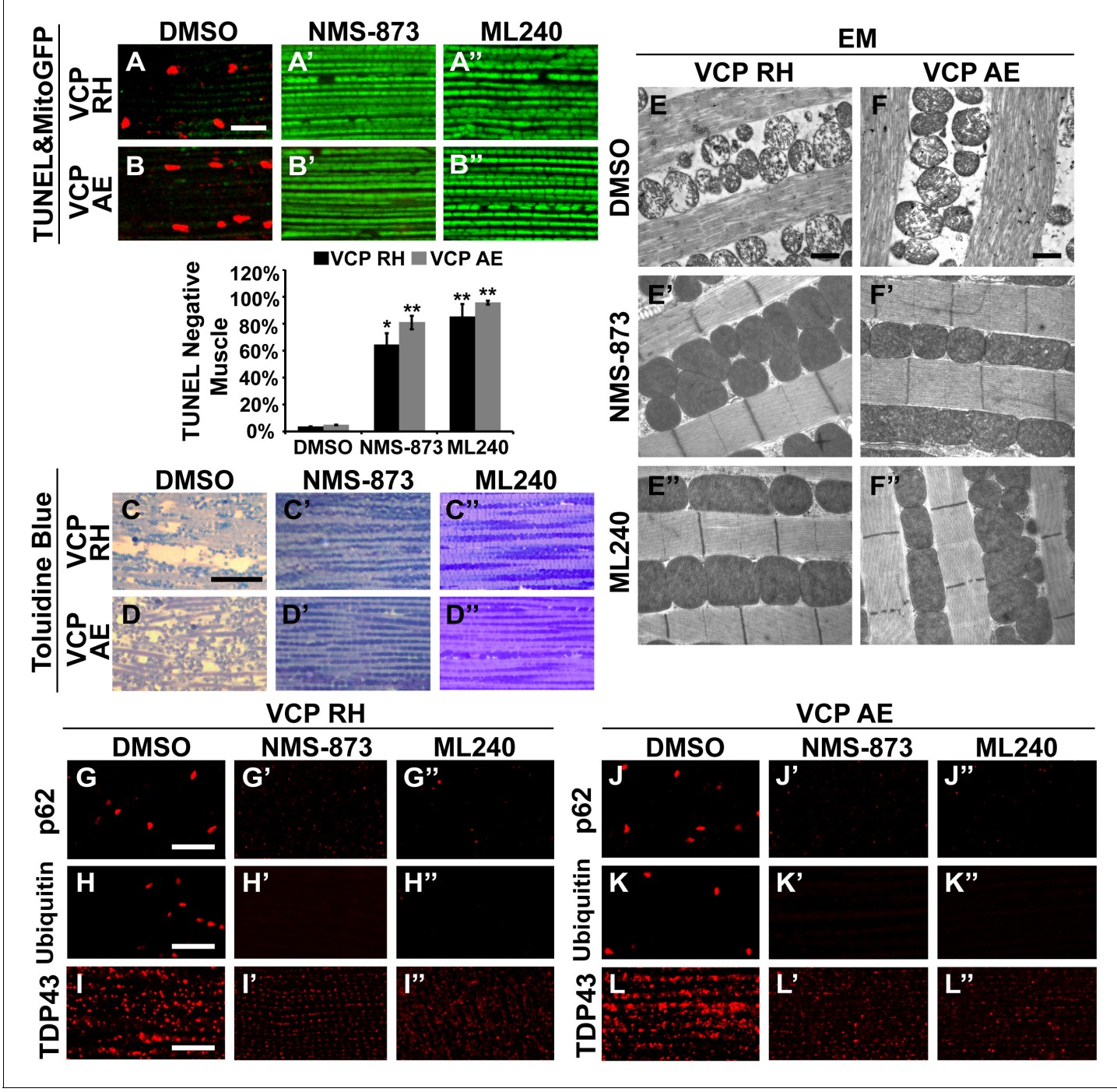

**Figure 8.** VCP inhibitors block mitochondrial defects, muscle tissue damage and muscle cell death in VCP disease mutant flies. (A–B'') Expression of VCP RH and AE leads to muscle cell death by 6 days after eclosion, as visualized through loss of MitoGFP and TUNEL positive signal in nuclei. In the DMSO feeding groups 3.4 ± 0.5% of VCP RH and 4.8 ± 0.6% of VCP AE flies are TUNEL negative. 30 μM NMS-873 or ML240 feeding since first instar larvae till 6 days after eclosion significantly blocks the muscle cell death, with 64.3 ± 17.0% (p=0.017) and 85.1 ± 19.0% (p=0.000) of VCP RH, and 80.9 ± 10.1% (p=0.002) and 95.6 ± 3.1% (p=0.000) of VCP AE muscles TUNEL negative. Data are shown as mean±½SD. Independent t test is used. Three independent rounds are performed, 15–20 flies are used each round. *p<0.05; **p<0.01. Scale bar: 10 μm. (C–D'') Toluidine Blue staining shows that 30 μM NMS-873 or ML240 feeding significantly rescue the disrupted muscle structure in VCP RH and AE flies. Scale bar: 40 μm. (E–F'') EM shows that 30 μM NMS-873 or ML240 feeding significantly rescue the broken myofibril structure and small mitochondria with broken cristae phenotypes in VCP RH and AE flies. Scale bar: 1 μm. Note that the mitochondria are still smaller than those from WT flies (*Figure 6B*). (G–L'') In 6-day-old flies, 30 μM NMS-873 (G'', H', J' and K') or ML240 (G'', H'', J'' and K'') treatments result in a significant decrease in Ref(2)P/p62 (G and J) and ubiquitin (H and K)

*Figure 8 continued on next page*

*Figure 8 continued*

aggregates in VCP RH and AE expressing flies; TDP43 mislocalization (I and L) is partially rescued as the large sarcoplasmic signal decreases, but no nuclear signal is observed (I', I'', L' and L''). Scale bar: 5 μm.

The following figure supplement is available for figure 8:

**Figure supplement 1.** VCP inhibitor feeding does not affect muscle viability and tissue integrity in 6-day-old flies.

increased by treatment with 250 nM ML240 for 6 hr. Overall mitochondrial morphology of IBMPFD fibroblasts does not change in response to inhibitor treatment (*Figure 9—figure supplement 1*). However, in a more sensitive assay in which rates of fusion are measured directly, mitochondrial fusion defects in IBMPFD patient fibroblasts were significantly rescued after 6 hr of treatment with 250 nM ML240 (*Figure 9B*). Finally, 50 nM and 100 nM ML240 treatment for 4 days also significantly enhanced the basal and maximal oxygen consumption rate (OCR) in patient fibroblasts (*Figure 9C*). Similarly, 1 nM and 10 nM NMS-873 treatment for 6 days significantly enhanced the maximal oxygen consumption rate (OCR) in patient fibroblasts as well (*Figure 9D*). The fact that two inhibitors with different mechanisms generate similar results in terms of rescuing IBMFPD pathologies reduces the likelihood that these are off-targeting effects. Together, these data strongly support the effects of VCP inhibitors as potential therapeutic tools for IBMPFD disease.

## Discussion

Missense mutations of VCP lead to the autosomal dominant disease IBMPFD. However, it is controversial whether these mutations cause disease through a dominant active or dominant negative manner. Moreover, the molecular mechanisms by which VCP mutations alter mitochondrial function are unclear. Here, we have investigated the endogenous function of VCP, cellular effects of expression of VCP disease mutants, and the ability of VCP inhibitors to suppress VCP disease pathology in *Drosophila* and patient fibroblasts. In summary, we have made the following findings:

Endogenous VCP regulates mitochondrial fusion (*Figure 10A*). Loss of VCP function leads to increased Mfn levels and increased mitochondrial size in multiple tissues. Conversely, VCP overexpression results in reduced Mfn levels and decreased mitochondrial size. VCP physically interacts with Mfn via domains that are also used to bind its substrates in other contexts. Thus, it is likely that Mfn is an endogenous VCP substrate. Here we provide the first *in vivo* evidence in muscle, the major disease tissue in IBMPFD, that Mfn is a specific endogenous target of VCP. Regulation of endogenous Mfn by VCP is unexpected, given that VCP is an abundant protein implicated in multiple cellular processes, but has not been shown to localize to mitochondria that are not depolarized by CCCP. We show that VCP-dependent removal of Mfn does not require Parkin or Mul1, E3 ligases known to ubiquitinate Mfn. How VCP brings about Parkin and Mul1-independent removal of Mfn requires further investigation.

IBMPFD is 100% penetrant, and myopathy is the most prevalent (affecting 90% of patients) and primary symptom of IBMPFD patients. We generated an IBMPFD disease model in adult *Drosophila* muscle, and used this system to monitor tissue pathology, and investigate the molecular mechanisms by which VCP disease mutants function. We show that compared to VCP WT, expression of VCP disease mutants results in tissue damage, mitochondrial fusion defects and decreased levels of Mfn (*Figure 10B*). These phenotypes are opposite to those associated with decreased levels of VCP, or expression of an ATPase defective mutant version of VCP. Expression of VCP disease mutants, but not a VCP ATPase defective mutant or VCP RNAi, also suppresses the mitochondrial defects mediated by accumulation of Mfn in *PINK1* null, *parkin* null, and *parkin mul1* double null mutants. Taken together, these results suggest that VCP disease mutants do not function as loss-of-function or dominant negative mutants, but rather as dominant activated mutants in regulating Mfn levels and bringing about mitochondrial fusion defects. This conclusion is further supported by our findings that mitochondrial fusion and Mfn levels are also decreased in VCP disease mutant fibroblasts.

Our finding that VCP overexpression suppresses a *parkin* null mutant in addition to a *PINK1* null stands in contrast to what was reported in Kim et al., who proposed that VCP-dependent degradation of Mfn required recruitment by Parkin. In addition, we also find that VCP overexpression

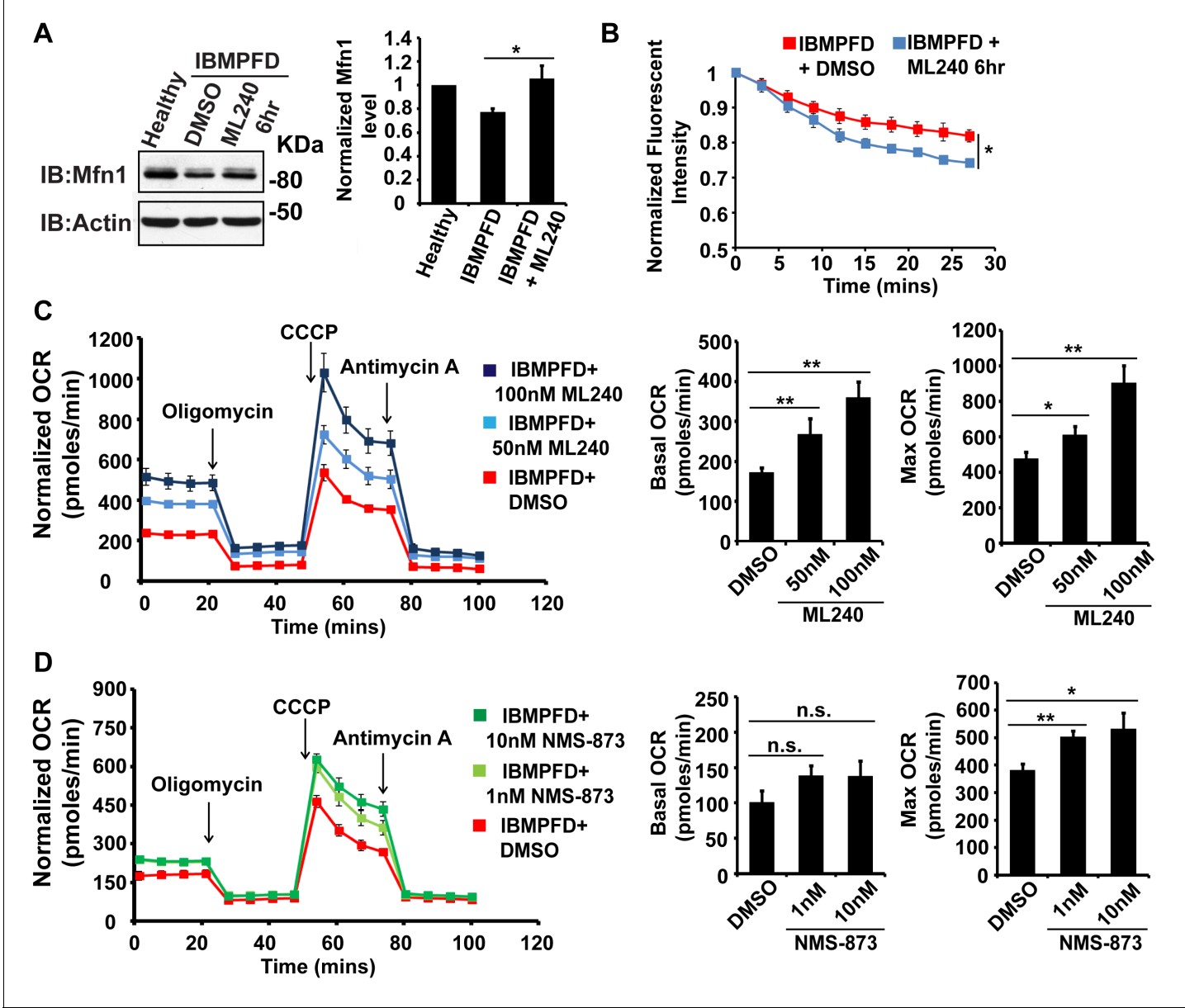

**Figure 9.** VCP inhibitor treatment significantly suppresses mitochondrial respiratory chain and fusion defects in $VCP^{R155H/+}$ IBMPFD patient fibroblasts. (A) After treatment of 250 nM ML240 for 6 hr, Mfn 1 level of $VCP^{R155H/+}$ IBMPFD patient fibroblasts is elevated from 0.77 ± 0.02 to 1.05 ± 0.11 (p=0.037, independent t test, N = 3), as compared with healthy controls, for which values are set as 1. (B) After treatment with 250 nM ML240 for 6 hr, the mitochondria fusion assay shows that the decreased mitochondrial fusion observed in IBMPFD patient's fibroblasts is significantly reversed (independent t test, *p<0.05). (C) 50 nM and 100 nM ML240 treatment for 4 days significantly increases basal (p=0.0001, DMSO v.s. 50 nM ML240; p=0.0027, DMSO v.s. 100 nM ML240, Welch's unpaired t test, N = 8) and maximal oxygen consumption rate (p=0.0415, DMSO v.s. 50 nM ML240; p=0.0028, DMSO v.s. 100 nM ML240, Welch's unpaired t test, N = 8) in the IBMPFD patient fibroblasts harboring $VCP^{R155H/+}$ mutation. (D) 1 nM and 10 nM NMS-873 treatment for 6 days significantly increases maximal oxygen consumption rate (p=0.0021, DMSO v.s. 1 nM NMS-873; p=0.0395, DMSO v.s. 10 nM NMS-873, N = 8), but not basal oxygen consumption rate (p=0.0994, DMSO v.s. 1 nM NMS-873; p=0.1804, DMSO v.s. 10 nM NMS-873, N = 8). Welch's unpaired t test is used. *p<0.05, **p<0.01, n.s, no statistical significance.

The following figure supplement is available for figure 9:

**Figure supplement 1.** ML240 and NMS-873 treatments do not significantly change the mitochondrial morphology in IBMPFD patients.

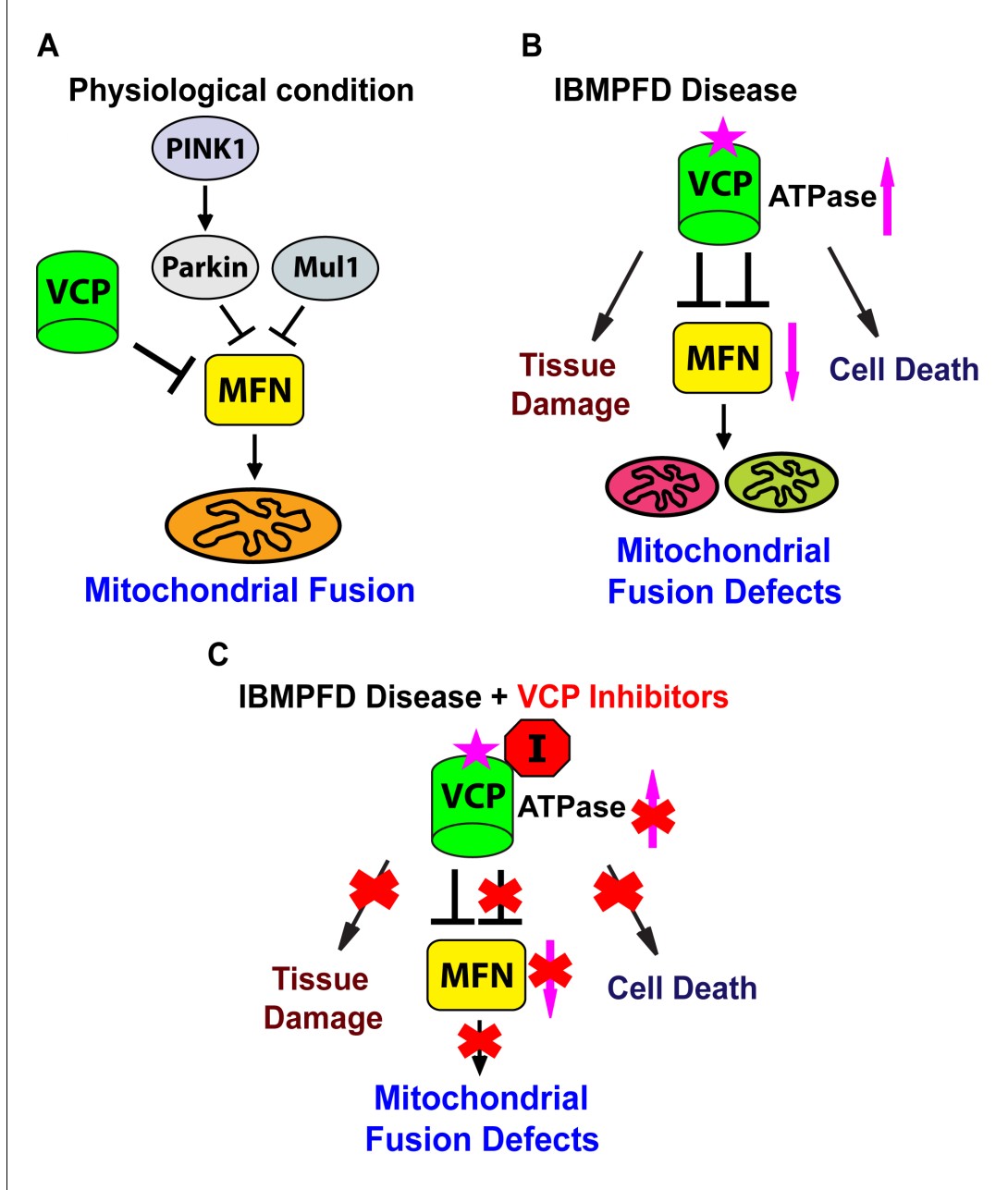

**Figure 10.** Proposed mechanisms of VCP disease mutants mediated mitochondrial defects and potential therapeutic role of VCP inhibitors. (**A**) Under physiological conditions, VCP, independent of PINK1/parkin and Mul1, negatively regulates Mfn protein levels, which are critical for the proper balance of mitochondrial fusion and fission. (**B**) In the IBMPFD disease, the enhanced ATPase activity of a VCP disease mutant protein results in excessive loss of Mfn and mitochondrial fusion, leading to mitochondrial fusion defects. Besides the mitochondrial defects, VCP disease mutants also generate pathology including adult muscle tissue damage and muscle cell death. (**C**) VCP ATPase inhibitors robustly relieve the pathology caused by VCP disease mutants associated with hyperactive VCP activity while leaving normal VCP-dependent functions intact.

suppresses phenotypes due to double mutants of *parkin* and *mul1*, which acts in a parallel pathway to degrade Mfn (*Yun et al., 2014*). Together, these observations show that VCP-dependent regulation of Mfn does not require *PINK1* or *parkin*. The VCP-dependent degradation of Mfn we observe is also specific and unlikely to be mediated through a more general process such as mitophagy, as levels of several other mitochondrial proteins are not altered by VCP overexpression. However,

these results do not exclude models in which Parkin activation further promotes VCP-dependent Mfn degradation, which may or may not involve mitophagy, under some conditions.

Regarding the mechanisms of VCP disease mutant action, *in vitro* biochemistry studies suggest that they have increased ATPase activities, thus favoring a dominant active model (*Weihl et al., 2006*; *Zhang et al., 2015*; *Niwa et al., 2012*; *Manno et al., 2010*; *Tang and Xia, 2013*). In addition, *in vivo* data from *Drosophila* shows that eye phenotypes caused by VCP disease mutant expression can be alleviated through loss of one copy of wildtype VCP. This also is consistent with models in which VCP mutants have increased activity (*Chang et al., 2011*). In contrast, in cell culture studies Bartolome, et al showed that VCP disease mutant expression led to mitochondria respiratory chain defects, as did VCP RNAi (*Bartolome et al., 2013*); Ritz, et al showed that VCP disease mutant expression impaired endocytosis to the same extent as that of VCP ATPase defective mutants (*Ritz et al., 2011*). While the above studies can be interpreted as showing that VCP disease mutants function as loss-of-function or dominant negative mutants, these results do not rule out the possibility that disease mutants are hyperactive alleles. For example, an alternative interpretation of the impairment of respiratory chain function observed following VCP disease mutant expression or following VCP RNAi is that either an increase or a decrease in the levels of VCP activity leads to mitochondrial defects, which is what we observed in this study.

Currently there is no treatment that can halt the progression of IBMPFD. Our findings that VCP disease mutants are hyperactive provide important therapeutic implications. Indeed, we find VCP inhibitors potently rescue multiple VCP disease phenotypes in flies and patient cells (*Figure 10C*). Importantly, suppression includes phenotypes beyond mitochondria (*Figure 10C*), involving TDP43, p62 and ubiquitin, suggesting that these inhibitors are likely to be effective in therapeutic settings. Of course, in order for VCP inhibitors to be useful as therapeutics, it will be necessary to inhibit mutant forms of VCP to an extent sufficient to suppress disease without bringing about a deleterious decrease in the activity of wildtype VCP. Our observation that *Drosophila* fed inhibitors are suppressed with respect to VCP disease phenotypes, but otherwise develop normally, suggests that this therapeutic goal may be possible. Finally, VCP disease mutants are associated with sporadic amyotrophic lateral sclerosis (*Abramzon et al., 2012*), and hereditary spastic paraplegia (*de Bot et al., 2012*) and Charcot-Marie-Tooth disease (*Gonzalez et al., 2014*). Our findings suggest possible therapeutic values of VCP inhibitors for these diseases.

## Materials and methods

### Molecular biology and constructs

Full length VCP (1-802aa), and truncated VCP cDNAs ΔN-VCP (186–802 aa), ΔN + D1-VCP (456–802 aa) and ΔD2-VCP (1–477 aa) were amplified from VCP cDNA derived from pUASt-VCP WT, a gift from Dr. Yun Nung Jan (*Rumpf et al., 2011*). All cDNAs were subcloned into pUASt vectors as translational fusions fused with the 6 Myc tag using the Gateway cloning system (Invitrogen). To generate pUASt-Mfn-3Flag, Mfn cDNA was obtained from an EST clone (*Drosophila* Genome Research Center, RE04414), and subcloned into pUASt with the 3Flag tag. To generate VCP RNAi 2, microRNA precursors targeting the coding region of VCP transcripts were cloned into the pUASt vector. The MicroRNA-based silencing technology has been described previously (*Chen et al., 2006*; *Ganguly et al., 2008*). All constructs generated above were verified with sequencing.

### *Drosophila* strains

UASt-VCP WT, VCP RH, AE and E2Q lines were gifts from Dr. Tzu Kang Sang (*Chang et al., 2011*). IFM-Gal4, UAS-Mfn RNAi, pCasper-DRP1-HA, PINK1[5], parkin[25], dpk[21], parkin[25]mul[A6] were described previously (*Clark et al., 2006*; *Deng et al., 2008*; *Yun et al., 2014*). VCP RNAi line1 flies were obtained from the Vienna *Drosophila* RNAi Center (VDRC 24354). VCP RNAi 2 was generated in the lab (see above). Efficiency of RNAi knocking down was verified by Western blot. 24B-Gal4, Mef2-Gal4 and EDTP-Gal4 were obtained from the Bloomington Stock Center at Indiana University. The pCasper-Mfn-HA flies (*Sandoval et al., 2014*) were a kind gift from Dr. Hugo Bellen at Baylor College of Medicine, Texas. The pCasper-Mfn-eGFP construct was a kind gift from Dr. C.K. Yao at Academia Sinica, Taipei. Flies carrying pCasper-Mfn-eGFP and pUASt-VCP RNAi two constructs were created through injection in a $w^{1118}$ genetic background (pCasper-Mfn-eGFP, X chromosome;

VCP RNAi line 2, second chromosome, Rainbow Transgenic Flies, Inc.). $vcp^{K15502}$, hsFLP22, neoFRT42D, and Ubi-mRFP.NLS, neoFRT42D were obtained from Bloomington stock center. *Drosophila* strains were maintained in a 25°C humidified incubator or at room temperature.

## Generation of germline nurse cell clones

$vcp^{K15502}$ was recombined with neoFRT42D. Female flies with the genotype: hsFLP22 / +; $vcp^{K15502}$, neoFRT42D / Ubi-mRFP.NLS, neoFRT42D or pCasper-Mfn-eGFP / hsFLP22; $vcp^{K15502}$, neoFRT42D / Ubi-mRFP.NLS, neoFRT42D were heat-shocked at 37°C for 2 hr every day from second instar larvae until pupae formation. After eclosion, flies were fed with dry yeast paste for 24 hr to stimulate oogenesis. The egg chambers then were dissected and fixed in 3.7% formaldehyde/Schneider's Buffer for immunoflurescence assays.

## TUNEL assay

Fly thoraxes of the relevant genotypes were cut and fixed in 4% paraformaldehyde/Schneider's Buffer at desired time points. Indirect flight muscles were then dissected out and separated. Muscles were further blocked with blocking buffer [50 mM Tris-Cl (pH 7.4), 0.1% Triton X-100, 188 mM NaCl] and assayed using the In Situ Cell Death Detection Kit (Roche). 15–20 fly thoraxes are dissected for each round. For each experiment, at least three rounds were performed.

## Toluidine blue staining and electronic microscopy

Fly thoraxes of the relevant genotypes were fixed in 1% paraformaldehyde/1% glutaraldehyde/ 0.1 M Phosphate Buffer, post-fixed in 1% osmium tetroxide/ddH20, dehydrated in gradient ethanol and embedded in Epon 812. After polymerization, sections were obtained using either glass knives or diamond knife (Diatome). 1.0–1.5 µm sections were stained with Toluidine blue. 80–90 nm sections were stained with uranyl acetate and lead citrate and examined by transmission electron microscope (UCLA Brain Research Institute Electron Microscopy Facility). At least 3 thoraxes of each genotype were examined.

## Mitochondrial cross-section size quantification

Electron microscopy images (10,000X magnification) were analyzed in Image J Software (National Institute of Health). After setting the scale, each mitochondrion on the image was selected and its area calculated. An independent t test was used to test for statistical significance. At least three images were analyzed for each thorax and at least 3 thoraxes of each genotype were examined.

## Immuofluoresence and confocal microscopy

For assays in indirect flight muscles, muscles of the corresponding age and genotypes were fixed in 4% paraformaldehyde/Schneider's Buffer, permeabilized in 0.3% Triton-X100/PBS. Rhodamine Phalloidin (Invitrogen, 1:500) was used for myofibrile staining. Muscle pieces were incubated with anti-TDP43 rabbit monoclonal antibody (ProteinTech, 1:100), anti-Ref(2)P/p62 rabbit polyclonal antibody (Abcam, 1:100), anti-P4D1 monoclonal antibody (ENZO, 1:100) at 4°C overnight. Washes were followed by Goat anti-rabbit/mouse Alexa Fluor 546 secondary antibody (Invitrogen 1:200) at room temperature for 2 hr. For assays of egg chambers, egg chambers were fixed in 3.7% formaldehyde/ Schneider's Buffer for 30 min and permeabilized with 0.4% Triton-X100/PBS for 4 hr. Egg chambers were incubated with anti-GFP rabbit polyclonal antibody (Invitorgen 1:100), anti-ATP5A mouse monoclonal antibody (Abcam, 1:100) at 4°C overnight and followed by goat anti-rabbit/mouse Alexa Fluor 488/546 secondary antibodies (Invitrogen 1:200) at 4°C overnight. Fibroblasts were fixed in 10% formalin for 10 min, 0.2% Triton-X100/PBS permeabilization for 15 min, 5% FBS/PBS blocking for 1 hr, anti-Tom20 mouse monoclonal antibody (BD, 1:200) at 4°C overnight and followed by goat anti-mouse Alexa Fluor 488 secondary antibodies (Invitrogen 1:200) at 4°C overnight. Images were taken using a LSM5 confocal microscope (Zeiss).

## S2 cell culture, transfection and dsRNA treatment

S2 cells were cultured in Schneider's Buffer, 10% FBS, 1% Penicillin and Streptomycin, 50 µM Tetracyclin at 25°C. Qiagen Effectene and was used as a transfection reagent according to the producer's instruction. Mfn DsRNA were designed according to a protocol from www.flyrrnai.org and

synthesized using the Mega T7 kit from Ambion. Cells were harvested 96 hr after dsRNA treatment. Mfn dsDNA-F: TGA GCA AAT ACC CCC AAA AG; Mfn dsDNA-R: GAT CTG GAG CGG TGA TTT GT.

## Fibroblast cell culture

Human primary fibroblasts from IBMPFD (GM21752) and age-matched control (GM00024) were obtained from the Coriell Institute for Medical Research (https://catalog.coriell.org) and grown in Dulbecco's Modified Eagle Medium (DMEM, Gibco) or Minimum Essential Media (MEM, Gibco) supplemented with 10% fetal bovine serum (FBS), 2 mM L-glutamine, and penicillin/streptomycin at 37°C and 5% CO2. Primary fibroblasts were immortalized by infection with retrovirus expressing hTERT (Addgene plasmid #1771). Infected cells were selected in puromycin (1.5 ug/mL) for 1 week and maintained in 1 ug/ml puromycin.

## Protein lysates and western blot

S2 cells, fly thoraxes or fibroblasts were lysed in RIPA Buffer with Protease Inhibitors (Roche) and 200 mM PMSF (Sigma). Protein lysates were centrifuged at 16,000g for 15 mins, and the supernatants boiled with 6XSDS sample Buffer (Bioland) at 95°C for 5 mins. Proteins were separated in SDS-PAGE Gels. Gels were transferred to PVDF membrane (Millipore) and incubated with primary antibody at 4°C overnight. Following several washes, blots were then incubated with secondary antibodies for 2 hr at room temperature and then washed extensively. Primary antibodies used include: Anti-HA mouse monoclonal antibody (Millipore, 1:1000), Anti-Myc mouse monoclonal antibody (Millipore,1:3000), Anti-Flag rabbit polyclonal antibody (Genscript, 1:2000), Anti-GFP rabbit polyclonal antibody (Invitrogen, 1:3000), Anti-Actin rabbit polyclonal antibody (Sigma, 1:2000), Anti-Tubulin mouse monoclonal antibody (Sigma, 1:4000), Anti-Porin mouse monoclonal antibody (Abcam, 1:2000), Anti-VCP rabbit monoclonal antibody (Cell Signalling Technology, 1:2000), Anti-Human Mfn 1 and 2 mouse monoclonal antibody (Abcam, 1:2000), Anti-MnSOD rabbit polyclonal antibody (Abcam, 1:2000) and Anti-NDUSF3 mouse monoclonal antibody (Abcam, 1:2000). Anti-*Drosophila* Mfn Rabbit polyclonal antibody (1:3000) was a gift from Dr. Alexander Whitworth (*Ziviani et al., 2010*). Donkey anti-mouse and anti-rabbit HRP conjugated secondary antibodies (GE Healthcare, 1:10000 or 20000) were used.

## Co-immunoprecipitation

48 hr after transfection, S2 cells were harvested and lysed in the RIPA Buffer with protease inhibitors. Protein lysates were incubated with Dynabeads G and primary antibody (Anti-Myc mouse monoclonal antibody, Millipore 1:300) at 4°C overnight and washed in 0.1%Tween/PBS four times and eluted in 2XSDS sample buffer (BioRad) and denatured at 95°C. The samples are assayed by western blot.

## *In vivo* and *in vitro* VCP inhibitors treatment

Powdered forms of NMS-873 (Selleckchem) and ML240 (Sigma-Aldrich) were dissolved in DMSO as stocks. Stock solution and DMSO as the vehicle control were diluted in ethanol/ddH2O and mixed with *Drosophila* food. Food dye was added to ensure thorough mix. Parents of desired genotypes were put in DMSO or inhibitors containing food for 3 days and removed. Thus, all growth following egg laying occurred in the presence of inhibitors. Immediately after eclosion, the progeny were transferred to newly prepared food vials containing the same concentration of DMSO or inhibitors. The progeny was assayed at the time points stated in the text. For human fibroblast treatment, stock solution of ML240, NMS-873 and DMSO vehicle control were diluted to the desired concentration and added to the culture media. Media were changed each day if the treatment was longer than 24 hr.

## Mitochondrial fusion assay in fibroblasts

The mitochondrial fusion assay was performed using photoactivatable mitoGFP (PA-GFP) localized to the mitochondrial matrix as previously described (*Mishra et al., 2014b*; *Karbowski et al., 2004*). Briefly, immortalized fibroblasts were infected with retroviruses expressing matrix-treated DsRed and matrix-targeted PA-GFP to create stable cell lines. Cells were plated on glass coverslips (Lab-Tek) and imaged live at 37°C on an LSM 710 confocal microscope (Carl Zeiss, Inc.). PA-GFP is

activated in a region of interest (5 μm x 5 μm) by illumination with a 405 nm laser. The activated signal is collected in z-stacks every 3 min over the next 30 min. Fusion events result in the dilution of the activated signal, and the average pixel intensity (for the entire cell) over time is a measurement of fusion rates (*Karbowski et al., 2004*). P-values are calculated using a Student's t-test on the slopes of intensity versus time for individual measurements (>20 per genotype).

## Mitochondrial respiration measurements

Oxygen consumption rates (OCR) were measured in immortalized fibroblasts using a Seahorse Biosciences Extracellular Flux Analyzer (Model XF96). Briefly, 10,0000 cells/well were plated the day before measurement into a 96 well culture plate. On the day of measurement, the media was exchanged for non-bicarbonate-containing DMEM (Sigma-Aldrich Cat. #D5030) media using the Seahorse PrepStation, and allowed to equilibrate for 1 hr at 37°C in room air prior to initiating measurements. Oxygen levels were measured over 5 min periods. Oxygen consumption rate was measured under basal and stress conditions as indicated. Oligomycin inhibits complex V and blocks ATP production dependent on respiration. CCCP uncouples oxidative phosphorylation and ATP production. It maximizes respiration without ATP synthesis. Antimycin A inhibits Complex III and thus blocks proton pumping and membrane potential formation. Drugs were injected automatically as indicated in the figure legends (Oligomycin 5 μM, CCCP 10 μM, Antimycin A 1 μM). For drug pre-treatment studies (*Figure 9C–D*), 5000 cells/well were plated into 96 well culture plates and pre-treated with ML240 for 4 days and NMS-873 for 6 days prior to OCR measurement. OCR values were normalized from total cellular content via sulforhodamine B measurements.

## Statistical analysis

The overall study design was a series of controlled laboratory experiments in *Drosophila* and human fibroblasts, as described in detail in the Figure legends sections. In all experiments, animals were randomly assigned to various experimental groups. The experiments were replicated at least three times (N was noted in each experiment) and the final analysis was presented. For in vivo experiments, 10–20 flies per group were used for each experiment. For the human fibroblasts experiments, details are described in the above methods sections. The data were analyzed using Statistical Package for the Social Sciences (SPSS) 16.0. The P values were assessed using a two-tailed unpaired Student's t test with P values considered significant as follows: *$p<0.05$ and **$p<0.01$. For seahorse assay in human fibroblasts, Welch's unpaired t test was used.

# Acknowledgements

We are grateful to the support of the Natalie R and Eugene S Jones Fund in Aging and Neurodegenerative Disease Research. We would like to thank Hansong Deng for first noting the suppression of *PINK1* mutant phenotype by VCP overexpression, Dr. Tzu Kang Sang from the National Tsing Hua University, Taiwan, for the VCP disease mutants allele flies, Dr. Hugo Bellen from the Baylor College of Medicine for the kind gift of pCasper-Mfn-HA flies, Dr. CK Yao from Academia Sinica, Taipei, for pCasper-Mfn-eGFP construct and Dr. Frank A Laski for the microtome usage.

# Additional information

## Funding

| Funder | Grant reference number | Author |
| --- | --- | --- |
| Ellison Medical Foundation | Senior Scholar award | Ming Guo<br>Bruce A. Hay |
| McKnight Endowment Fund for Neuroscience | | Ming Guo |
| Glenn Family Foundation | | Ming Guo |
| National Institutes of Health | NIA R01 | Ming Guo |
| National Institutes of Health | NINDS EUREKA award | Ming Guo |

Natalie R. and Eugene S. Jones Fund in Aging and Neurodegenerative Disease Research Ming Guo

The funders had no role in study design, data collection and interpretation, or the decision to submit the work for publication.

## Author contributions

TZ, PM, BAH, Conceptualization, Methodology, Writing—original draft, Writing—review and editing; DC, Conceptualization, Supervision, Data curation, Formal analysis, Writing—original draft, Writing—review and editing; MG, Conceptualization, Supervision, Funding acquisition, Data curation, Formal analysis, Methodology, Writing—original draft, Writing—review and editing

## Author ORCIDs

Ting Zhang, http://orcid.org/0000-0002-9636-6554

Prashant Mishra, http://orcid.org/0000-0003-2223-1742

David Chan, http://orcid.org/0000-0002-0191-2154

Ming Guo, http://orcid.org/0000-0002-1889-4271

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
