## [Decision Letter]

Thank you for submitting your article "VCP inhibitors relieve Mitofusin-dependent mitochondrial defects due to VCP disease mutants" for consideration by *eLife*. Your article has been favorably evaluated by K VijayRaghavan (Senior Editor and Reviewer #3) and three reviewers, one of whom, Hugo Bellen, is a member of our Board of Reviewing Editors.

The reviewers have discussed the reviews with one another and the Reviewing Editor has drafted this decision to help you prepare a revised submission.

Summary:

In the current manuscript, Zhang et al. explore the role of VCP in mitochondrial dynamics as well as the associated pathogenesis in IBMPFD (inclusion body myopathy, Paget's disease of the bone with frontotemporal dementia). VCP is a type II ATPase that has predicted functions in many processes such as mitochondrial function, cell cycle regulation, DNA repair, ERAD, endolysosomal sorting, autophagy, and organelle biogenesis. This manuscript shows that overexpression of VCP in *Drosophila* indirect flight muscles resulted in smaller mitochondria whereas VCP RNAi caused mitochondria to elongate. Similarly, mitochondria in nurse cells with loss of function VCP alleles are more elongated or clumped. Muscle lysates with higher VCP levels displayed less Mfn expression and Nurse cells with loss of function VCP alleles showed increased Mfn levels. All these results are consistent with prior reports in mammals and flies that VCP mediates degradation of mitofusins. The authors do correct an apparent error in the literature by showing VCP OE rescues muscle phenotypes caused by both PINK1 and Parkin loss. In general, the paper is well written, the figures are nice, and the data are compelling. However, some data sets are not novel, and the reader has a hard time distinguishing what is novel and what is not. In addition, a number of important issues remain to be addressed.

1) Overexpression of VCP mutants (RH and AE) caused p62 aggregates and ubiquitin inclusions and mitochondrial defects. Correlating with one aspect of VCP mutant expression, Mfn mutation caused similar mitochondrial defects. Do Mfn mutant flies display p62 aggregates and ubiquitin inclusions too?

2) In Figure 5—figure supplement 2 the authors show that mild Mfn overexpression suppresses the mitochondrial fragmentation caused by VCP RH and AE expression. Does Mfn overexpression rescue the TUNEL, Toluidine Blue, TDP43, p62 and ubiquitin phenotypes shown in Figure 4 to be caused by VCP RH and AE expression? One main unresolved mystery is whether or not the VCP mutation-induced defects that are rescued by VCP inhibition (Figure 7 and Figure 8) arise from the Mfn decrease and subsequent mitochondrial defects or from effects on other VCP involved pathways.

3) Is the conclusion that the small effects on Mfn levels seen in patient cells in Figure 6 causes or contributes to IBMPFD pathophysiology?

4) Do Mfn haploinsufficient mice or flies (or people) display phenotypes?

5) Does increasing Mfn levels in IBMPFD cells rescue the respiratory defects noted in Figure 6?

6) Do the VCP inhibitors cause mitochondria to elongate in the IBMPFD patient cells?

7) A second main undetermined issue is whether or not the VCP inhibitors cause their phenotypic rescue effects by inhibiting VCP or through other off target activities. A mutant VCP that is functional but insensitive to the inhibitors would be a rigorous way to explore that important issue.

---

## [Author Response]

*Summary:*

*In the current manuscript, Zhang et al. explore the role of VCP in mitochondrial dynamics as well as the associated pathogenesis in IBMPFD (inclusion body myopathy, Paget's disease of the bone with frontotemporal dementia). VCP is a type II ATPase that has predicted functions in many processes such as mitochondrial function, cell cycle regulation, DNA repair, ERAD, endolysosomal sorting, autophagy, and organelle biogenesis. This manuscript shows that overexpression of VCP in Drosophila indirect flight muscles resulted in smaller mitochondria whereas VCP RNAi caused mitochondria to elongate. Similarly, mitochondria in Nurse cells with loss of function VCP alleles are more elongated or clumped. Muscle lysates with higher VCP levels displayed less Mfn expression and Nurse cells with loss of function VCP alleles showed increased Mfn levels. All these results are consistent with prior reports in mammals and flies that VCP mediates degradation of mitofusins. The authors do correct an apparent error in the literature by showing VCP OE rescues muscle phenotypes caused by both PINK1 and Parkin loss. In general, the paper is well written, the figures are nice, and the data are compelling. However, some data sets are not novel, and the reader has a hard time distinguishing what is novel and what is not. In addition, a number of important issues remain to be addressed.*

We thank the reviewers for suggesting that we distinguish what is novel and what is not. Here are the novel findings of the paper:

1) While some groups have reported the effects of VCP overexpression on Mfn levels, we report here that loss of endogenous VCP in *Drosophila* nurse cells leads to an increase in the levels of endogenous Mfn. This is important as VCP is a highly abundant protein with many targets. To demonstrate that endogenous VCP regulates Mfn in vivo is highly significant.

2) We have shown that overexpression of VCP suppresses parkin null mutant phenotypes in addition to PINK1 null mutants. This alters the interpretation of how VCP overexpression functions to regulate PINK1/parkin related pathway.

3) No prior report suggests that VCP disease mutants suppress Mfn in a dominant active fashion.

4) No prior report suggests that human disease fibroblasts exhibit mitochondrial fusion defects that are dependent on Mfn.

5) No prior report suggests that VCP inhibitors reverse the pathology related to disease models or human disease fibroblasts.

We are grateful that the reviewers made this suggestion and have incorporated these points in the manuscript.

*1) Overexpression of VCP mutants (RH and AE) caused p62 aggregates and ubiquitin inclusions and mitochondrial defects. Correlating with one aspect of VCP mutant expression, Mfn mutation caused similar mitochondrial defects. Do Mfn mutant flies display p62 aggregates and ubiquitin inclusions too?*

This is a very interesting question and we thank reviewers for suggesting this. We performed the experiments and the results are now incorporated in Figure 4. As mfn loss-of-function mutants are homozygous lethal, we examined phenotypes due to muscle-specific mfn knockdown. These flies show TDP mislocalization (Figure 4’), tissue disintegration (Figure 4’), p62 and ubiquitin accumulation (Figure 4), as with the VCP IBMPFD mutants.

*2) In Figure 5—figure supplement 2 the authors show that mild Mfn overexpression suppresses the mitochondrial fragmentation caused by VCP RH and AE expression. Does Mfn overexpression rescue the TUNEL, Toluidine Blue, TDP43, p62 and ubiquitin phenotypes shown in Figure 4 to be caused by VCP RH and AE expression? One main unresolved mystery is whether or not the VCP mutation-induced defects that are rescued by VCP inhibition (Figure 7 and Figure 8) arise from the Mfn decrease and subsequent mitochondrial defects or from effects on other VCP involved pathways.*

This is an interesting question, and we performed the experiments and added new figure panels in the Supplementary figure. We have shown that mild (2-fold) increases of Mfn levels using the genomic rescue flies, while rescuing mitochondrial fragmentation cause by VCP disease mutant RH and AE expression at day 2 (Figure 5—figure supplement 2, does not rescue muscle tissue integrity, muscle cell death, TDP43 mislocalization, p62 and ubiquitin aggregates seen at day 6 (Figure 5—figure supplement 3). However, it is important to note that we are unable to fully address whether overexpression of mfn can suppress phenotypes due to VCP RH and AE expression, since mfn overexpression by itself shows strong phenotypes – TUNEL positivity and tissue disintegration assays by Toluidine Blue (Yun et al., 2014).

Together, our observations suggest that VCP regulates endogenous levels of Mfn, and that a 2-fold increase of Mfn levels can suppress mitochondrial abnormalities. While we cannot ascertain, for technical reason, whether VCP inhibition rescues VCP IBMPFD pathology solely due to a decrease in Mfn, these results argue that downregulation of Mfn is at least an important contributor to IBMPFD muscle pathology.

*3) Is the conclusion that the small effects on Mfn levels seen in patient cells in Figure 6 causes or contributes to IBMPFD pathophysiology?*

We speculate that the Mfn level decrease and mitochondrial fusion defects probably contribute to IBMPFD pathology as one of multiple pathogenesis pathways, given that VCP is involved in multiple other pathways. As stated immediately above, we cannot fully address this.

*4) Do Mfn haploinsufficient mice or flies (or people) display phenotypes?*

In *Drosophila*, homozygous Mfn loss-of-function flies (*mfn^B^*) are 3^rd^ instar laval lethal [Sandovol, et al., *eLife*, 2014,PMID: 25313867]. *mfn^B^/+* heterozygous flies do not display muscle tissue damage or mitochondrial morphology defects as compared to wildtype (data not shown). In mice, skeletal muscle from *mfn1* and *mfn2* tissue-specific double knockout mice display decreased muscle volume, and mitochondrial fragmentation with abnormal cristae. But *mfn1-/- mfn2+/-* and *mfn1+/- mfn2-/-* mice grow to normal size and skeletal muscle mitochondria have mild structural changes [Chen, et al., Cell, 2010, PMID: 20403324]. In humans, no case on Mfn haploinsufficiency has been reported.

*5) Does increasing Mfn levels in IBMPFD cells rescue the respiratory defects noted in Figure 6?*

This is a very interesting question. However, we are unable to address it unambiguously. This is because Mfn1 or 2 overexpression results in toxic effects on human cells. As reported extensively in the literature (see below), Mfn1 or 2 overexpression results in perinuclear clustering of fragmented mitochondria, [Santel et al; Huang et al]. Functionally, Mfn1 overexpression leads to decreased mitochondrial ATP generation, membrane potential and decreased mitochondrial motility [Park et al., 2008; Park et al., 2012]; Mfn2 overexpression leads to significant loss of mitochondrial membrane potential and cell apoptosis [Huang et al]. Mfn1 overexpression increases cell’s vulnerability under stress conditions [Park et al., 2014]. We now note in the text that this question remains outstanding.

1 Santel, A., et al., Mitofusin-1 protein is a generally expressed mediator of mitochondrial fusion in mammalian cells.Journal of cell science, 2003. 116(Pt 13): p. 2763-74.

2 Huang, P., T. Yu, and Y. Yoon, Mitochondrial clustering induced by overexpression of the mitochondrial fusion protein Mfn2 causes mitochondrial dysfunction and cell death. European journal of cell biology, 2007. 86(6): p. 289-302.

3 Park, K.S., et al., Selective actions of mitochondrial fission/fusion genes on metabolism-secretion coupling in insulin-releasing cells.The Journal of biological chemistry, 2008. 283(48): p. 33347-56.

4 Park, K.S., A. Wiederkehr, and C.B. Wollheim, Defective mitochondrial function and motility due to mitofusin 1 overexpression in insulin secreting cells. The Korean journal of physiology & pharmacology: official journal of the Korean Physiological Society and the Korean Society of Pharmacology, 2012. 16(1): p. 71-7.

5 Park, Y.Y., et al., MARCH5-mediated quality control on acetylated Mfn1 facilitates mitochondrial homeostasis and cell survival. Cell death & disease, 2014. 5: p. e1172.

*6) Do the VCP inhibitors cause mitochondria to elongate in the IBMPFD patient cells?*

To address reviewer’s comments, we performed the mitochondrial morphological assay and include the new Figure 9—figure supplement 1. We do not see mitochondrial morphological changes. However, in a more sensitive assay, which measures the rate of mitochondria fusion directly, the decreased frequency of fusion observed in IBMPFD patient cells is significantly suppressed in the presence of a VCP inhibitor. This was our original Figure 9. We now note more explicitly in the text these points.

*7) A second main undetermined issue is whether or not the VCP inhibitors cause their phenotypic rescue effects by inhibiting VCP or through other off target activities. A mutant VCP that is functional but insensitive to the inhibitors would be a rigorous way to explore that important issue.*

This point could be tested if there were VCP mutants that did not affect ATPase activity while still being resistant to NMS-873 and ML240. Unfortunately, no such mutants have been reported to date. Importantly, the experiments we report here are carried out with two VCP inhibitors that function through distinct mechanisms. NMS-873 inhibits VCP function through allosteric change of the protein structure [Magnaghi et al., Nat Chem Biol, 2013, PMID: 23892893], while ML240 function through competitive inhibition of ATP binding [Chou et al., ChemMedChem, 2013, PMID: 23316025]. The fact that two inhibitors with different mechanisms generate similar results in terms of rescuing IBMFPD pathologies suggests these are unlikely to be off-targeting effects.